# Functional screening for anti-CMV biologics identifies a broadly neutralizing epitope of an essential envelope protein

Thomas J. Gardner[1], Kathryn R. Stein[1], J. Andrew Duty[1,2], Toni M. Schwarz[1], Vanessa M. Noriega[1], Thomas Kraus[2], Thomas M. Moran[1,2] & Domenico Tortorella[1]

The prototypic β-herpesvirus human cytomegalovirus (CMV) establishes life-long persistence within its human host. The CMV envelope consists of various protein complexes that enable wide viral tropism. More specifically, the glycoprotein complex gH/gL/gO (gH-trimer) is required for infection of all cell types, while the gH/gL/UL128/130/131a (gH-pentamer) complex imparts specificity in infecting epithelial, endothelial and myeloid cells. Here we utilize state-of-the-art robotics and a high-throughput neutralization assay to screen and identify monoclonal antibodies (mAbs) targeting the gH glycoproteins that display broad-spectrum properties to inhibit virus infection and dissemination. Subsequent biochemical characterization reveals that the mAbs bind to gH-trimer and gH-pentamer complexes and identify the antibodies' epitope as an 'antigenic hot spot' critical for virus entry. The mAbs inhibit CMV infection at a post-attachment step by interacting with a highly conserved central alpha helix-rich domain. The platform described here provides the framework for development of effective CMV biologics and vaccine design strategies.

[1] Department of Microbiology, Icahn School of Medicine at Mount Sinai, New York, New York 10029, USA. [2] Center for Therapeutic Antibody Development, Icahn School of Medicine at Mount Sinai, New York, New York 10029, USA. Correspondence and requests for materials should be addressed to D.T. (email: Domenico.tortorella@mssm.edu).

Human cytomegalovirus (CMV) is a betaherpesvirus with a seroprevalence of 60–90% that can cause morbidity and mortality in susceptible individuals[1,2]. At particular risk for CMV disease are newborns, organ transplant recipients, AIDS patients and the elderly[3]. CMV exhibits a broad cell tropism and utilizes multiple envelope protein complexes to attach and fuse with host cell membranes. There is limited structural information available for the fusogenic CMV glycoprotein(s) comprised of gB, gH and gL; and their functional interactions are poorly understood[4–7]. gB catalyses membrane fusion during viral entry, and gH and gL likely serve as factors which activate gB to permit pH-independent fusion at the cellular membrane or within macropinocytic vesicles[7–9]. gH and gL are present in the trimeric gH/gL/gO complex that contributes to viral entry into diverse cell types[8,10–12]. The pentameric complex (PC) also contains gH/gL together with three additional proteins UL128, UL130 and UL131a. The PC is required for viral entry into epithelial, endothelial and myeloid cells where the virion enters through endocytosis or macropinocytosis followed by a low pH-dependent fusion event[9,13–15].

Existing CMV drugs exhibit some efficacy in treating infection, although toxicity, drug–drug interactions and the development of drug-resistant viral strains are common limitations[16]. Immunotherapeutics, in contrast, offer a safe strategy for prevention or mitigation of CMV infection including foetal transmission. CMV-hyperimmune globulin has proven somewhat effective for transplant patients[17–19] and treatment of congenital CMV infection[20–23], though it may not completely protect the CMV-vulnerable foetus[24]. A number of drawbacks exist for immunoglobulin preparations, including batch-to-batch variability, low concentrations of neutralizing antibodies, as well as risks associated with blood-derived products[25]. gB-specific antibodies can impart some protection from infection[26–29], and recent work has revealed that monoclonal antibodies (mAbs) which target the PC[30–32] are only effective at preventing CMV infection of non-fibroblast cells in vitro[31,33,34].

In this study, we identified various novel CMV mAbs that block virus infection. Utilizing the high-throughput capabilities of automated hybridoma colony picking coupled with a high-throughput neutralization (HTN) assay, we generated a panel of potent, broad-spectrum neutralizing mAbs targeting conformational epitopes of gH. These mAbs revealed a highly conserved domain on the CMV-gH protein that is susceptible to neutralization, thus validating this rapid and effective framework for the development of potent and broadly specific antiviral biologics.

## Results

**Generation of neutralizing α-CMV mAbs.** Our goal was to generate broad-spectrum α-CMV mAbs using a HTN assay[35,36]. Neutralization of the reporter CMV strain AD169 encoding for a chimeric IE2 protein coupled to yellow fluorescent protein (YFP; AD169$_{IE2-YFP}$) was examined using human CMV-positive serum, α-gH mAb 14-4b and controls (CMV-negative serum, α-MHC class I heavy chain (HC-10) and α-CD44). Fluorescence intensity from infected MRC5 fibroblasts was measured with a fluorescent cytometer[35]. Using cells infected with mock-treated virus as 100% infection, only the α-gH mAb (14-4b) and CMV-positive serum neutralized a CMV infection providing proof-of-concept for the AD169$_{IE2-YFP}$-neutralization assay (Fig. 1a).

We hypothesized that antibodies targeting the core binding and fusion envelope proteins (for example, gM/gN, gB and gH/gL) would be ideal targets for broad-spectrum mAbs. To that end, mice were immunized with AD169 because it does not incorporate an intact pentamer complex in the virion[13] and thus would not elicit a humoral response to the immunodominant proteins UL128, UL130 and UL131a (refs 29,31). A HTN assay using serum from the immunized mice (Fig. 1b) identified the animals with the highest neutralizing capacity and ∼2,000 hybridoma clones were generated from two animals and subjected to the AD169$_{IE2-YFP}$-HTN assay (Supplementary Fig. 1). Of the clones that decreased infection by >50%, six clones neutralized infection in a dose-dependent manner. These clones were subjected to a 5-point dose-dependence HTN assay and demonstrated the capacity to neutralize infection (Fig. 1c). To determine whether the neutralizing clones recognize infected cells, AD169$_{IE2-YFP}$-infected fibroblasts were subjected to flow cytometry with the hybridoma clones following assay optimization with the α-gH mAb 14-4b (Fig. 1d,e). The neutralizing hybridoma clones exclusively bound to AD169$_{IE2-YFP}$-infected cells (Fig. 1e) suggesting they target viral envelope proteins.

**CMV-neutralizing mAbs target envelope proteins gB and gH.** What are the protein target(s) of the CMV-neutralizing antibodies? Proteins recovered from metabolically labelled AD169$_{IE2-YFP}$-infected cells following immunoprecipitation with the neutralizing mAbs were resolved by SDS–polyacrylamide gel electrophoresis (SDS–PAGE; Fig. 2a). The neutralizing mAbs recovered polypeptides segregating into two distinct groups. Group I mAbs recovered three major protein species (∼130, 100 and 60 kDa (Fig. 2a, left panel)), a result consistent with immature and mature forms of gB[37]. The neutralizing mAbs 10C10 and 5C3 (group II) recovered distinct proteins (∼100, 37 and 12 kDa (Fig. 2a, right panel)): a pattern consistent with gH, gL and UL128 proteins. This would be expected because although AD169 virions do not incorporate an intact PC due to a truncated UL131 protein[38–40], UL128 is expressed in the endoplasmic reticulum (ER) of infected cells. Additionally, the absence of UL130 from an immunoprecipitation targeting a gH complex is consistent with previous findings[13]. Further validation that the mAbs targeted glycoproteins was based on the faster migration pattern upon peptide-N-glycosidase F (PNGase) digestion (Fig. 2b). The predominant polypeptides recovered by the neutralizing mAbs 2F4 (group I) and 10C10 (group II) from infected fibroblasts were confirmed by mass spectrometry analysis to be gB, gH and gL, respectively (Supplementary Table 1). Collectively, the neutralizing mAbs recognized the core envelope proteins gB and gH/gL.

To confirm the exact protein target of the neutralizing mAbs, U373 cells expressing gH or gB (Fig. 2c) were subjected to flow cytometry (Fig. 2d) or immunoprecipitation studies (Fig. 2e). An α-gB and α-gH immunoblot confirmed expression of the envelope proteins (Fig. 2c, lanes 1–2 and 3–4). Following the establishment of the flow cytometry assay (Supplementary Fig. 2), group I mAbs 2F4, 5A6, 7H7 and 8H2 bound exclusively to U373$^{gB}$ cells (Fig. 2d, left column), while the α-gH antibody 14-4b, and the group II antibodies 5C3 and 10C10 bound only to U373$^{gH}$ cells (Fig. 2d, right column). The mAbs' specificity for gB or gH was validated by immunoprecipitation (Fig. 2e). Taken together with the inability of the mAbs to bind denatured protein in an immunoblot assay (Supplementary Fig. 2), the collective set of data indicates that the neutralizing mAbs bind to conformational epitopes of CMV gB and gH.

**CMV-neutralizing mAbs selectivity inhibit virus infection.** To examine the specificity of the CMV-neutralizing antibodies, we utilized a CMV TB40/E strain that expresses FLAG-YFP (TB40/E$_{FLAG-YFP}$) in a HTN assay[36,41] (Fig. 3a). The α-gH mAbs 14-4b, 10C10 and 5C3 significantly blocked infection; yet, the

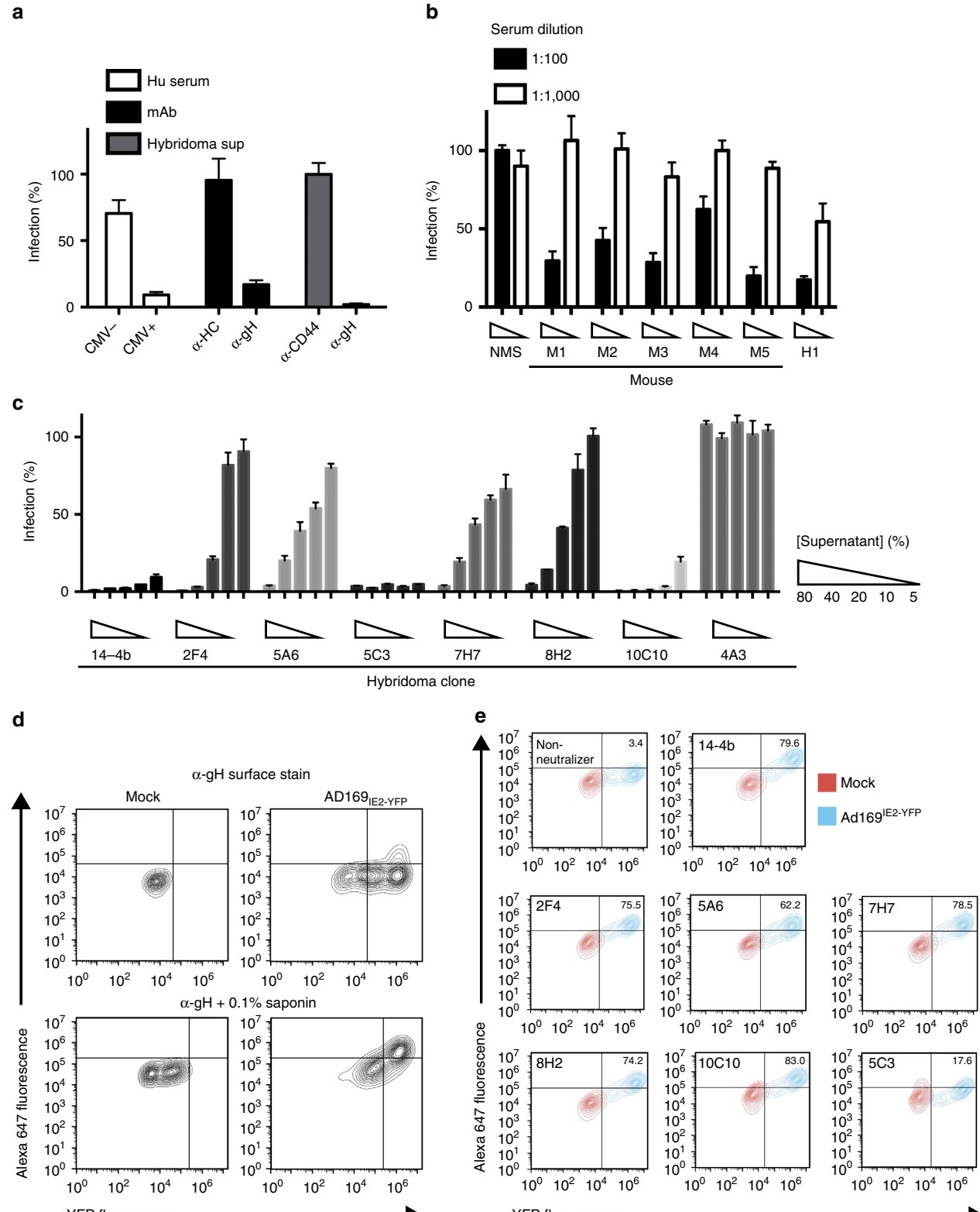

**Figure 1 | Identification of CMV-neutralizing mAbs using a HTN assay.** (**a**) Human serum, purified mAb (anti-MHCI heavy chain (HC) mAb (HC-10), anti-gH mAb 14-4b) and hybridoma supernatant were tested in a HTN assay for their ability to block infection of fibroblasts by AD169$_{IE2-YFP}$. (**b**) Serum from mice inoculated with AD169 was tested for its ability to neutralize AD169$_{IE2-YFP}$ infection at 1:100 (black bars) or 1:1000 (white bars). Normal mouse serum (NMS) or CMV+ human serum (H1) were used as controls. (**c**) Hybridoma supernatant from 6 CMV-neutralizing clones was tested in 5-point dilutions (5–80%) for their ability to inhibit AD169$_{IE2-YFP}$ infection. Supernatant from the neutralizing anti-gH mAb 14-4b and supernatant from the non-neutralizing hybridoma clone 4A3 were utilized as controls. (**d**) MRC5 cells infected with AD169$_{IE2-YFP}$ were exposed to anti-gH (14-4b) flow cytometry staining without (top row) or with (bottom row) permeabilization with saponin. (**e**) Mock-infected MRC5 cells (red) or infected with AD169$_{IE2-YFP}$ (blue) were permeabilized and then exposed to hybridoma supernatant from the CMV-neutralizing clones followed by detection with flow cytometry. Experiments for (**a–c**) were performed in technical triplicate and s.d. is depicted.

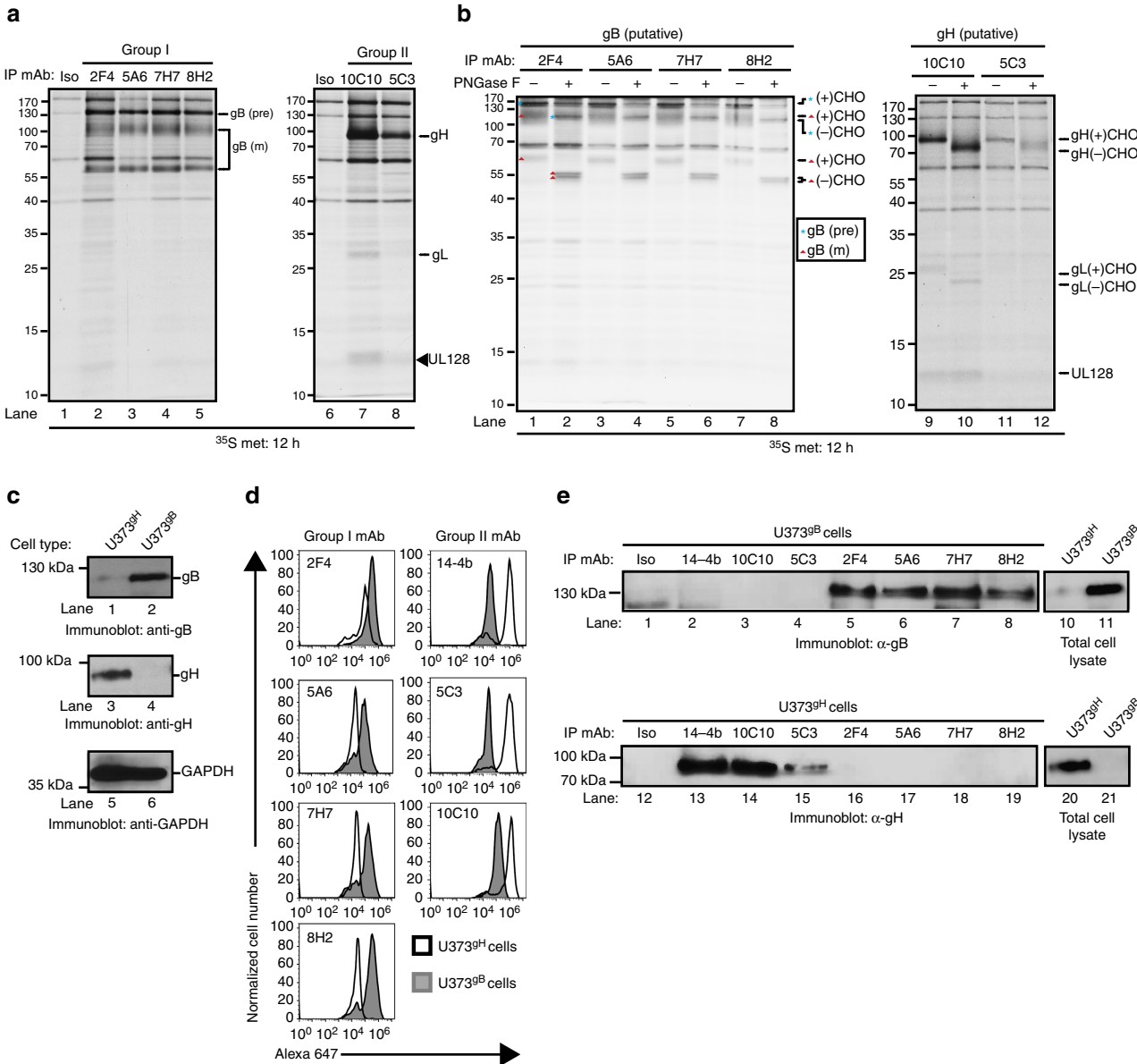

**Figure 2 | Identification of CMV-neutralizing mAb targets** (**a**) Lysates from metabolically labelled AD169$_{IE2-YFP}$-infected MRC5 cells were exposed to the CMV-neutralizing mAbs and recovered immune complexes were resolved by SDS–PAGE. Arrows denote the suspected peptide identity. (**b**) Immune complexes recovered from the metabolically labelled cell lysates were exposed to PNGase treatment and resolved by SDS–PAGE. Arrows denote the suspected peptide identity and glycosylation state. (**c**) Total cell lysates from U373 glioblastoma cells stably expressing gH (U373$^{gH}$) or gB (U373$^{gB}$) were resolved by SDS–PAGE and exposed to immunoblot for gB (lanes 1–2), gH (lanes 3–4) and GAPDH (lanes 5–6) proteins. (**d**) U373$^{gH}$ (white peaks) and U373$^{gB}$ (grey peaks) cells were permeabilized and exposed to the neutralizing CMV mAbs followed by detection with flow cytometry. mAb 14-4b was used as a positive control for labelling of the gH protein. (**e**) Total cell lysates from U373$^{gB}$ (lanes 1–8) and U373$^{gH}$ (lanes 12–19) were exposed to the CMV-neutralizing mAbs. The recovered immune complexes were resolved by SDS–PAGE and exposed to anti-gB (lanes 1–8) or anti-gH (lanes 12–19) antibody. Total cell lysates from U373$^{gB}$ (lanes 10–11) and U373$^{gH}$ (lanes 20–21) validated expression of the respective CMV protein in each cell type. Relative molecular mass markers are indicated in all relevant figures.

α-gB mAbs did not reduce infection levels implying that the α-gB mAbs displayed selective inhibition of CMV strains. To quantify the potency of the neutralizing mAbs, IC50 values were determined using AD169$_{IE2-YFP}$ or TB40/E$_{FLAG-YFP}$-HTN assays (Fig. 3b,c). The neutralizing mAbs potently inhibited an AD169$_{IE2-YFP}$ infection (IC50 values: 0.24 µg ml$^{-1}$ (5C3) to 4.2 µg ml$^{-1}$ (7H7)), while only the α-gH mAbs 10C10 and 5C3 neutralized TB40/E$_{FLAG-YFP}$ (IC50 values: 0.38 µg ml$^{-1}$ and 0.07 µg ml$^{-1}$ (Fig. 3b)). Consistent with Fig. 3a, the purified α-gB mAbs were unable to reduce infection levels of TB40/E$_{FLAG-YFP}$ (Fig. 3c), despite recognition of CMV gB by

the α-gB mAbs as demonstrated by immunoprecipitation studies from TB40/E WT-infected cells (Fig. 3d). The data demonstrate that the α-gH mAbs are capable of neutralizing multiple CMV strains.

**α-gH mAbs block virus infection and dissemination**. To determine the broad-spectrum neutralization capacity of α-gH mAbs, we developed a HTN assay using a fluorescently labelled α-Immediate Early (IE) mAb (α-IE$^{Alexa488}$) (Fig. 4a). AD169-WT, TB40/E, VHL/E and TR strains pre-incubated with α-gH mAbs or Cytogam, a polyclonal IgG preparation from

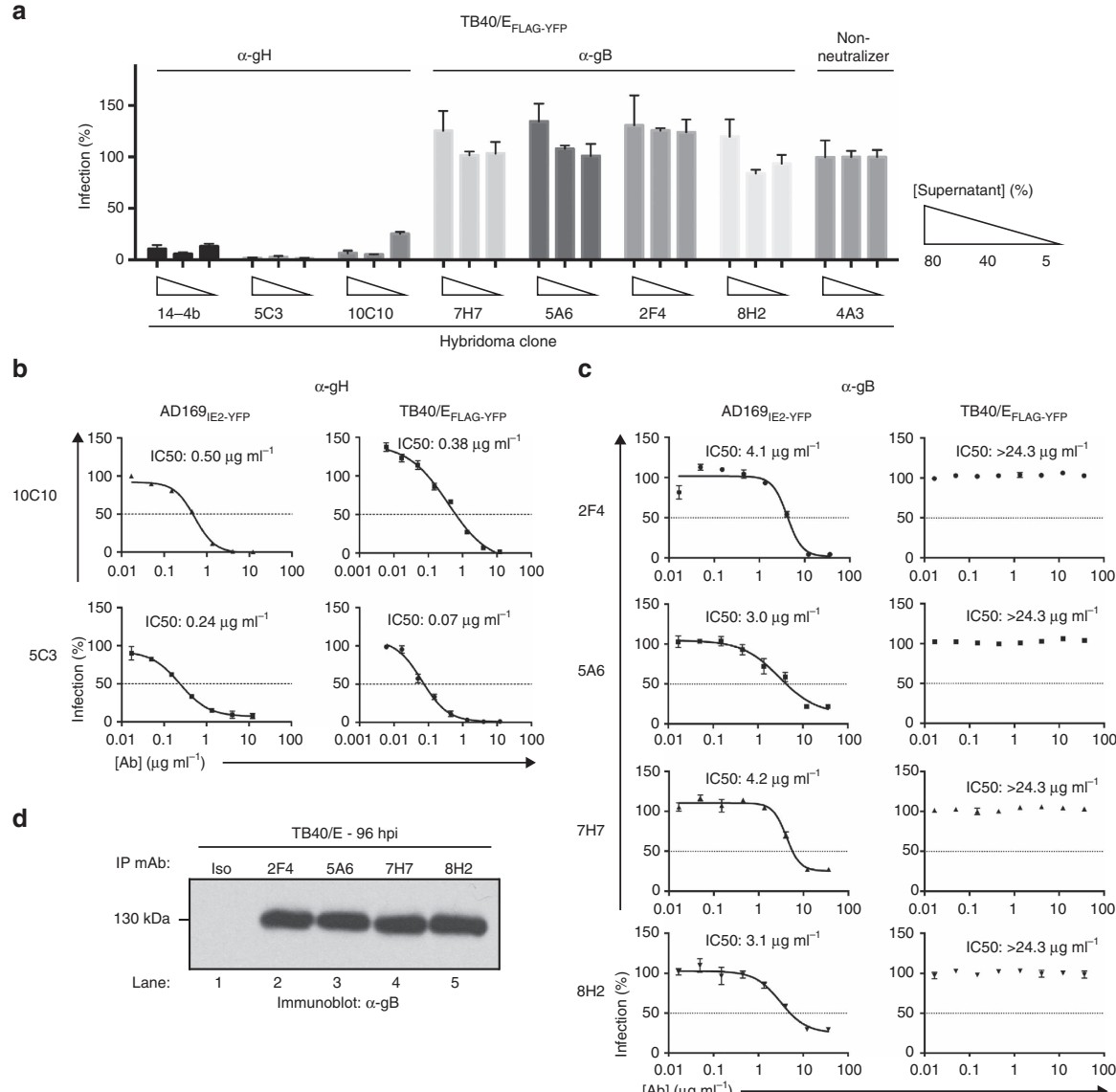

**Figure 3 | Examination of the neutralizing capacity of anti-CMV mAbs.** (**a**) Hybridoma supernatant from the 6 CMV-neutralizing mAbs was tested at 3 concentrations (5, 40 and 80%) for their ability to inhibit TB40/E$_{FLAG-YFP}$ infection of MRC5 cells. Supernatant from the neutralizing anti-gH mAb 14-4b and supernatant from the non-neutralizing hybridoma clone 4A3 were utilized as controls. Experiments were performed in technical triplicates and s.d. is depicted. (**b**) Anti-gH mAbs 10C10 and 5C3 were pre-incubated with AD169$_{IE2-YFP}$ (left panel) and TB40/E$_{FLAG-YFP}$ (right panel) at 8 concentrations (0.01–12 μg ml$^{-1}$) and infection levels of MRC5 cells was subsequently measured. (**c**) Anti-gB mAbs 2F4, 5A6, 7H7 and 8H2 were pre-incubated with AD169$_{IE2-YFP}$ (left panel) and TB40/E$_{FLAG-YFP}$ (right panel) at 8 concentrations (0.01–12 μg ml$^{-1}$) and infection levels of MRC5 cells was subsequently measured. Non-linear regression analysis was performed and the half maximal inhibitory concentration (IC50) was calculated for all antibodies. (**d**) MRC5 cells infected with TB40/E were harvested at 96 hpi and total cell lysates were exposed to the anti-gB antibodies. Recovered immune complexes were resolved by SDS–PAGE and exposed to anti-gB immunoblot (lanes 1–5). Relative molecular mass markers are indicated. Experiments for (**a**–**c**) were performed in technical triplicate and s.d. is depicted. Respective virus neutralization experiments were replicated four times each.

CMV seropositive individuals[42], were analyzed for virus infection in MRC5 fibroblasts. Both 10C10 and 5C3 demonstrated enhanced neutralization of all strains (IC50: 0.28–4.18 μg ml$^{-1}$) when compared to Cytogam, which was ineffective at blocking virus infection (Fig. 4a and Supplementary Table 2). Additionally, analysis of TB40/E, DAVIS, VHL/E and TR neutralization by 10C10 and 5C3 via immunoblot analysis demonstrated complete reduction of Late Antigen expression (Supplementary Fig. 3).

Potential variation in affinity of the fluorescent α-IE antibody between virus strains may complicate the comparison of IC50 values. Thus, traditional plaque reduction assays were performed in MRC5 cells infected with clinical-like strains

DAVIS, TB40/E, VHL/E and TR (Fig. 4b). The α-gH mAbs reduced plaque numbers with IC50 values between 0.02 and 2.43 μg ml$^{-1}$, whereas Cytogam was unable to limit plaque numbers of the clinical-like strains with the exception of VHL/E (IC50: 0.12 μg ml$^{-1}$; Fig. 4b and Supplementary Table 2). These results demonstrate the effectiveness of α-gH neutralizing mAbs in blocking virus infection.

**α-gH mAbs broadly inhibit infection of non-fibroblast cells.** To further analyse the efficacy of the α-gH mAbs as broad-spectrum antivirals, we examined their potency in blocking virus infection of human retinal pigment epithelial cells (ARPE-19) using

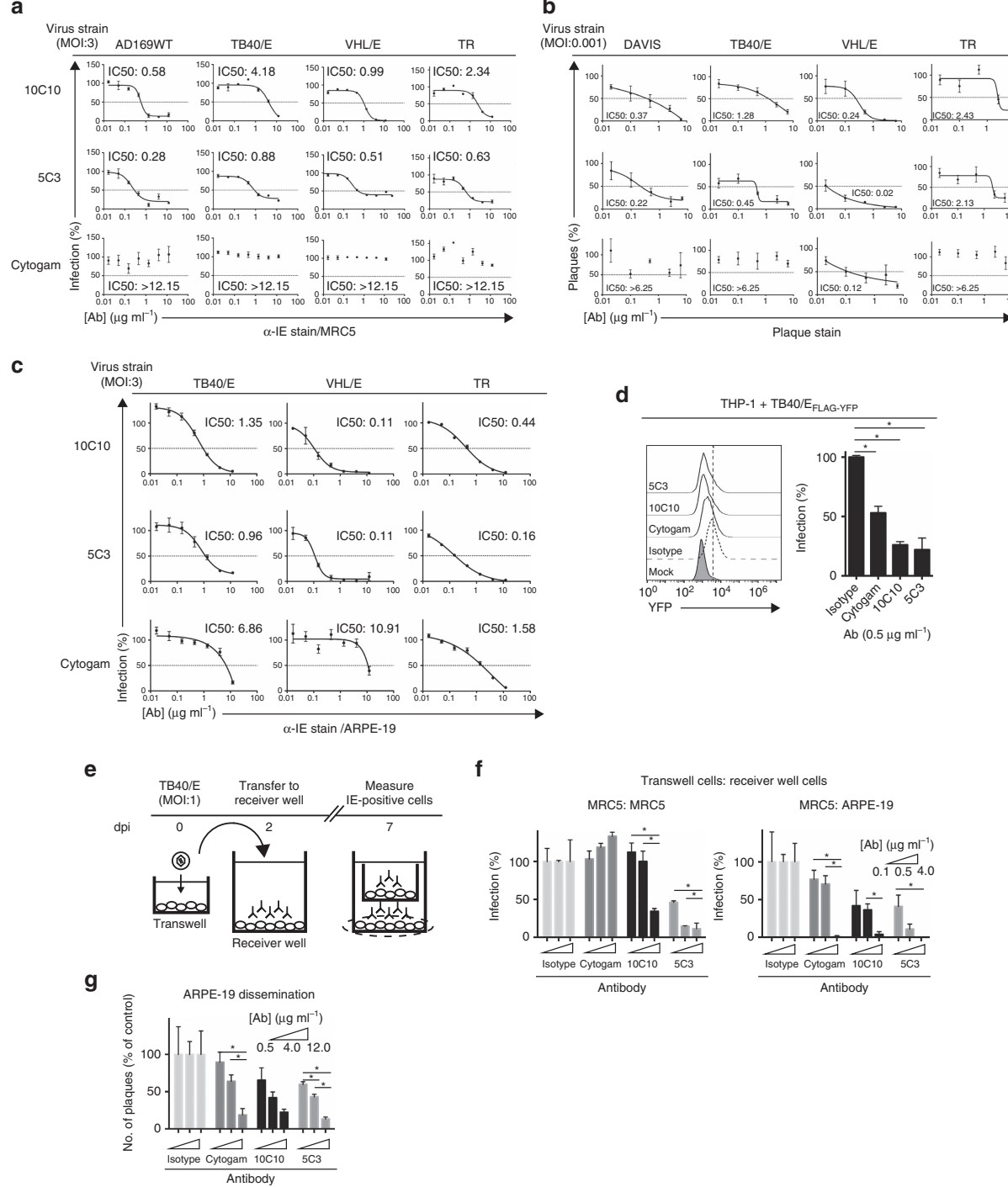

**Figure 4 | Examination of the potency and versatility of anti-gH mAbs in epithelial cells.** (**a**) Cytogam and mAbs 10C10 and 5C3 were pre-incubated with CMV strains AD169, TB40/E, VHL/E and TR (0.01–12 µg ml$^{-1}$), and subsequent infection levels of MRC5 cells was analyzed by immunostaining with anti-Immediate Early (IE) gene product (α-IE$^{Alexa488}$) antibody. Experiments were performed in technical triplicate. (**b**) Cytogam and mAbs 10C10 and 5C3 were pre-incubated with CMV strains DAVIS, TB40/E, VHL/E and TR (0.025–6.25 µg ml$^{-1}$), and subsequent infection levels of MRC5 cells was analyzed by plaque assay. Data represents averages from three experiments performed in duplicate. (**c**) Cytogam and mAbs 10C10 and 5C3 were pre-incubated with CMV strains TB40/E, VHL/E and TR (0.01–12 µg ml$^{-1}$), and subsequent infection levels of human retinal pigment epithelial cells (ARPE-19) was measured by anti-IE staining. Experiments were performed in technical triplicate. Non-linear regression analysis was performed and the half maximal inhibitory concentration (IC50) was calculated for all antibodies under all conditions. (**d**) Cytogam and mAbs 10C10 and 5C3 were pre-incubated with the fluorescent CMV strain TB40/E$_{FLAG-YFP}$ (0.5 µg ml$^{-1}$) before infection of THP-1 cells (MOI: 3) and fluorescence levels were analyzed by flow cytometry at 4 dpi. Experiments were performed in technical triplicate. (**e**) MRC5 cells were seeded on a transwell insert with 3 µm pore and infected (MOI:1) with TB40/E. At 2 dpi the transwell insert was transferred to a receiver well containing an isotype control, Cytogam or mAbs 10C10 and 5C3 (0.1–4 µg ml$^{-1}$) and at 7 dpi the cells from the receiver layer were analyzed by α-IE$^{Alexa488}$ immunostain. (**f**) MRC5 (left panel) and ARPE-19 (right panel) cells from the transwell infection experiment were analyzed for infection levels by α-IE$^{Alexa488}$ immunostain. Experiments were performed in technical triplicate. (**g**) Infection levels of ARPE-19 cells infected with TB40/E (MOI:0.1) and exposed to an isotype control, Cytogam or mAbs 10C10 and 5C3 (0.5–12 µg ml$^{-1}$) over a 10 day period were analyzed by α-IE$^{Alexa488}$ immunostain. Experiments were performed in technical triplicate. S.d. is depicted for all experiments. *$P < 0.05$ (Student's two-tailed $t$ test).

TB40/E$_{FLAG-YFP}$ (Supplementary Fig. 3). The mAbs 10C10 and 5C3 inhibited infection with IC50 values of 0.13 µg ml$^{-1}$ (10C10) and 0.04 µg ml$^{-1}$ (5C3) (Supplementary Table 2). Additionally, 10C10 and 5C3 inhibited infection of TB40/E, VHL/E and TR with IC50 values from 0.11 to 1.35 µg ml$^{-1}$ (Fig. 4c, top 2 rows and Supplementary Table 2) and Cytogam exhibited variable neutralizing capacity with IC50 values of > 6 µg ml$^{-1}$ for TB40/E and VHL/E infection, and 1.58 µg ml$^{-1}$ for the clinical strain TR (Fig. 4c bottom row and Supplementary Table 2). Finally, 10C10 and 5C3 (0.5 µg ml$^{-1}$) reduced infection by > 50% with Cytogam decreasing infection by ~ 50% of TB40/E$_{FLAG-YFP}$-infected monocytes (THP-1; Fig. 4d). We conclude that the α-gH mAbs are effective neutralizing mAbs of various viral strains and cell types.

**α-gH neutralizing mAbs block viral dissemination**. To assess the capability of the α-gH mAbs to block nascent infection, we next examined the ability of the mAbs to limit cell-free virus spread in a transwell assay (Fig. 4e). TB40/E-infected fibroblasts in the top chamber of a transwell dish were positioned above a receiver well of MRC5 fibroblasts or ARPE-19 epithelial cells containing α-gH mAbs, Cytogam, or an isotype control (0.1–4 µg ml$^{-1}$) followed by analysis of virus infection 7 days post-infection (dpi; Fig. 4f). The α-gH mAbs decreased infection in a dose-dependent manner in both cell types with an enhanced inhibition in ARPE-19 cells, whereas Cytogam prevented virus infection of only ARPE-19 cells. Further, we examined the effectiveness of the α-gH mAbs to block viral dissemination through multiple infection cycles in a monolayer. ARPE-19 cells infected with TB40/E (MOI: 0.1) were treated with α-gH mAb, Cytogam, or an isotype control (0.5–12 µg ml$^{-1}$) at 48 h post infection (hpi) and analyzed for virus-infected cells at 10 dpi (Fig. 4g). The α-gH mAbs and Cytogam limited virus dissemination based on the decreased number of IE-positive plaques. Additionally, infection of primary CD14 + monocytes followed by coculture with MRC5 cells in the presence of the α-gH mAbs demonstrated their ability to block dissemination from fibroblasts to monocytes (Supplementary Fig. 4).

To further characterize the efficacy of the anti-gH antibodies 5C3 and 10C10 to limit viral dissemination, AD169$_{IE2-YFP}$-infected (MOI: 0.1) MRC5 cells were incubated with isotype, Cytogam, 5C3 and 10C10 at 24 hpi without replenishing the antibodies and analyzed for cytopathogenicity and viral plaques (Supplementary Fig. 5). Analysis of the infected cells indicated that 5C3 and 10C10 were more effective at limiting cytopathic effects, plaque number and the number of IE2-YFP-positive cells/plaque. Collectively, our data support the model that the α-gH mAbs are quite effective at limiting viral dissemination.

**α-gH mAbs bind to multiple envelope protein complexes**. To determine if the α-gH mAbs recognize both the gH/gL-trimer and gH/gL-pentamer complexes, proteins recovered by 10C10 and 5C3 from $^{35}$S-methionine-labelled, TB40/E-infected cells were subjected to PNGase treatment and resolved by non-reducing SDS–PAGE (Fig. 5a). Based on the relative molecular weights of the proteins, we observed large complexes ( > 250 kDa) likely consisting of covalently linked multimers of glycosylated gH/gL/gO trimers[43], ~ 130 kDa species comprised of the gH/gL/UL128 trimer, ~ 100–130 kDa proteins made of gH/gL dimers and polypeptides < 100 kDa that represent the nascent gH protein (Fig. 5a, lanes 3 and 5). PNGase treatment confirmed the glycosylation status of the recovered proteins (Fig. 5a, lanes 4 and 6). These data were validated by immunoblot analysis for gH, gL and UL128 (Fig. 5b). The detection of UL128 covalently bound to gH/gL was used as a proxy for the PC. As expected, gH and gL

exist within diverse complexes ranging from ~ 90 kDa to > 250 kDa (Fig. 5b, lanes 1–3 and 4–6) and the gH/gL/UL128 species, which migrated at ~ 130 kDa (Fig. 5b, lanes 7–9). Further, U373 cells expressing gH/gL/hemagglutinin (HA)-tagged gO (U373$^{gH/gL/gO-HA}$) or gH/gL/UL128 (U373$^{gH/gL/UL128}$) were used to validate 10C10 and 5C3 binding to UL128- (Fig. 5c, lanes 1–3, 13–15 and 19–21) and gO-containing gH/gL complexes (Fig. 5c, lanes 4–6, 10–12 and 16–18). Additionally, mass spectrometry analysis of the viral proteins recovered from TB40/E-infected cells (72 hpi) with the anti-gH mAb 10C10 identified gH, gL, gO, UL128, UL130 and UL131a (Supplementary Table 3). These findings further support the model that the anti-gH mAbs recover both the gH/gL-trimer and pentamer complexes. Additional biochemical studies defined that 5C3 binds preferentially to the gH/gL complex, whereas 10C10 bound equally well to gH or gH/gL (Supplementary Fig. 6).

**10C10 and 5C3 block a post-attachment entry step**. Which step of the entry process do mAbs 10C10 and 5C3 block? To address this question, AD169$_{IE2-YFP}$ was pre-incubated with 10C10, 5C3, Cytogam or an isotype control to allow for viral binding by the mAbs, and added to MRC5 fibroblasts (pre-attachment; Fig. 6). Alternatively, AD169$_{IE2-YFP}$ was added to MRC5 cells at 4 °C to prevent virus internalization, followed by addition of 10C10, 5C3, Cytogam and an isotype control (post-attachment); then returned to 37 °C. Virus infection was measured by YFP-positive cells and normalized to isotype-treated cells. The mAbs 10C10 and 5C3 blocked infection pre- and post-viral attachment suggesting inhibition of a step downstream of viral binding.

**Epitope mapping of 5C3 and 10C10**. A binding competition assay was performed to define whether 10C10 and 5C3 recognize the same epitope (Fig. 7). U373$^{gH/gL}$ cells incubated with a fluorescent-labelled Alexa647 10C10 antibody (10C10$^{647}$, 1 µg ml$^{-1}$) and competed with non-fluorescent 10C10, 5C3 or an isotype control (1–20 µg ml$^{-1}$; Fig. 7a, top row) were analyzed by flow cytometry. Only incubation with 10C10 resulted in a dose-dependent fluorescent decrease indicating displacement of 10C10$^{647}$ (centre panel). Intriguingly, in competition experiments with a fluorescent-labelled 5C3 antibody (5C3$^{647}$; Fig. 7a, bottom row), both 5C3 and 10C10 caused a decrease in 5C3$^{647}$ binding, although a more pronounced reduction was observed following incubation with 5C3 (compare center and right panels). Quantification of percentage fluorescence change revealed that 10C10 (black bars) only displaces 5C3$^{647}$ at high mAb concentrations (Fig. 7b). These data suggest that 10C10 and 5C3 bind to a similar region and 10C10 may have a slightly higher affinity for the gH polypeptide.

Truncation analysis revealed that 5C3 recognized an epitope between residues 295–717 and 10C10 binds to residues 132–717 (Supplementary Fig. 7). To further define the epitope of the mAbs, we utilized a cyclic peptide microarray platform (PEPperCHIP) consisting of overlapping cyclic peptides of 7, 10 and 13aa derived from gH (Fig. 8a). Remarkably, 5C3 bound consistently to a peptide (RREIFIVET) corresponding to domain 2 of the gH (aa. 485–493) protein[44] (Fig. 8b). A 3D model of CMV-gH based on the crystal structure of HSV-1 gH (ref. 45) predicted that aa481–493 lies within an alpha helix-rich domain on the surface of CMV-gH (Fig. 8c).

To confirm the identity of the 5C3 epitope and determine if 10C10 also targeted this domain, a flow cytometry-based binding assay was established using gH alanine mutants. Assay conditions were initially established using HEK-293 cells expressing wild-type-gH, gH$^{478-79AA}$ (a mutant substituting two alanines upstream of the putative epitope region (aa478–79), and

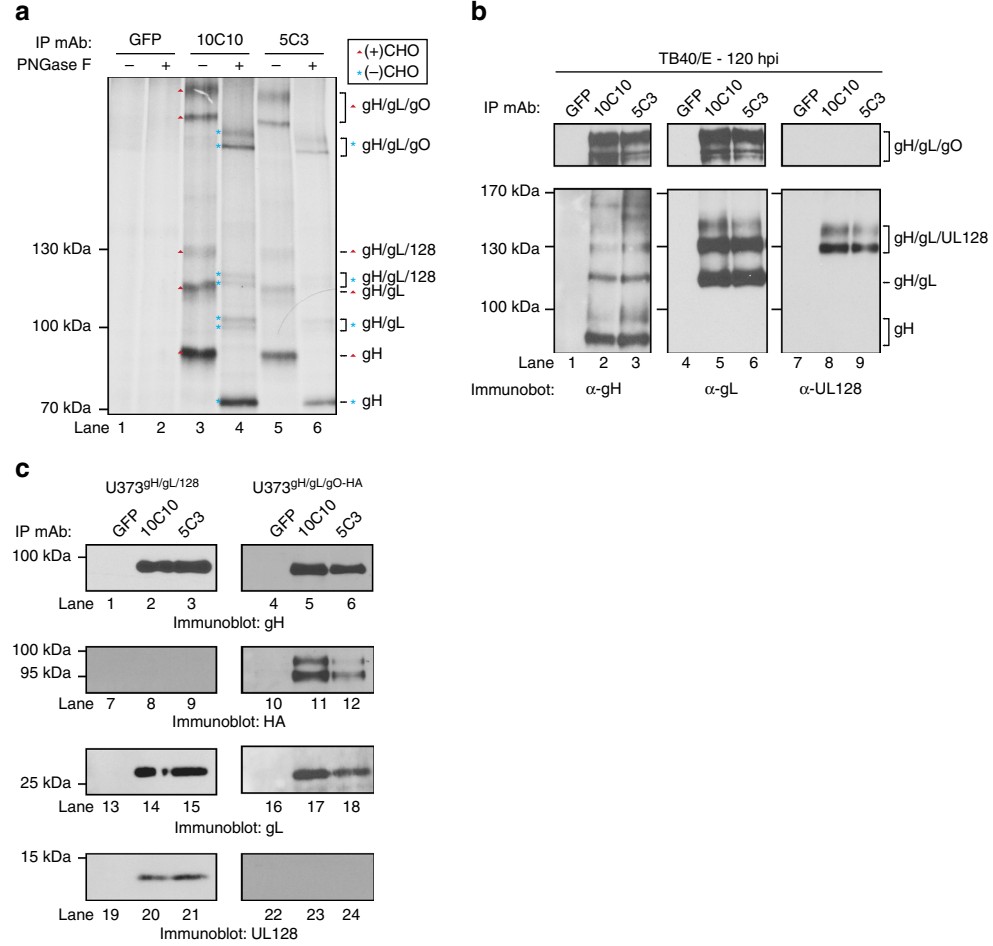

**Figure 5 | α-gH mAbs bind to multiple glycoprotein complexes** (**a**) Lysates from metabolically labelled TB40/E-infected MRC5 cells were exposed to an isotype control mAb recognizing GFP, or mAb 10C10 and 5C3. Recovered immune complexes were split and treated with PNGase F, then resolved by non-reducing SDS–PAGE. Arrows denote the identity of glycosylated protein complexes and asterisks denote the identity of de-glycgosylated protein complexes. (**b**) Lysates from TB40/E-infected MRC5 cells were exposed to a GFP mAb, 10C10 or 5C3. Immune complexes were resolved by non-reducing SDS–PAGE followed by immunoblot for gH (lanes 1–3), gL (lanes 4–6) and UL128 (lanes 7–9). Lines denote the glycoprotein complexes. (**c**) Lysates from U373 cells stably expressing gH/gL/UL128 (U373^gH/gL/UL128) and gH/gL/gO-HA (U373^gH/gL/gO-HA) were exposed to GFP, 10C10 or 5C3 mAbs and the recovered immune complexes were resolved by SDS–PAGE and subjected to immunoblot for gH (lanes 1–6), HA (lanes 7–12), gL (lanes 13–18) and UL128 (lanes 19–24).

gH$^{W168A}$ (an alanine point mutant that disrupts α-gH mAb MSL-109 binding[46] and used as a control; Fig. 8d and Supplementary Table 4). 10C10, 5C3 and MSL-109 recognized wild-type gH and gH$^{478-79AA}$; whereas, MSL-109 as expected, weakly bound to gH$^{W168A}$ (Fig. 8d). Interestingly, the slight decrease in 10C10 and 5C3 binding to gH$^{W168A}$ demonstrates the sensitivity of these antibodies to conformational changes of gH.

Next, two mutants with alanine substitutions spanning the putative epitope region were generated for binding analysis (Fig. 8e–g and Supplementary Table 4). Binding of MSL-109 to both constructs validated the intact conformation of gH (Fig. 8e). Remarkably, 10C10 and 5C3 were unable to bind to the gH$^{480-A-486}$ mutant; yet 5C3 binding was restored for the gH$^{485-A-492}$ mutant and not for 10C10 (Fig. 8e,f). For higher-resolution mapping of the epitope, 10C10 or 5C3 binding to additional alanine mutants (gH$^{480AA481}$, gH$^{482AA483}$, gH$^{484AA485}$, gH$^{487AA488}$, gH$^{489AA490}$ and gH$^{491AA492}$) was performed (Fig. 8g and Supplementary Table 4). While MSL-109 and 5C3 bound to all of the mutants, 10C10 binding was completely abrogated on disruption of amino acids 484–487. These data demonstrate that the epitopes of 5C3 and 10C10 are contained within residues 481-HTTERREIFI-490. This region is

highly conserved (100% identity) among diverse CMV strains (Fig. 8h) defining the mAbs' broad specificity.

Notably, a recent study implicated gH residues 485–493 and 671–675 as critical for binding of the anti-gH mAb 14-4b (ref. 47). To investigate the requirement of residues 485–493 for binding by 14-4b, the gH alanine mutants were evaluated for recognition by this mAb (Supplementary Fig. 8). Strikingly, aa's 487–488 were more critical for interaction with 14-4b than residues 484–485 based on the abrogation of antibody binding. Interestingly, the 14-4b binding to gH mutants was more similar to the binding profile of 10C10, yet the neutralization data was in line with the IC50 values of 5C3 (Supplementary Fig. 8). Further, competition studies using 10C10 and 5C3 conjugated with a fluorescent probe (Alexa-647) and non-labelled 14-4b for binding to U373gH/gL cells demonstrated that 14-4b does not compete with 10C10$^{647}$ binding and only limits 5C3$^{647}$ binding at higher 14-4b concentrations. These findings suggest that the three anti-gH antibodies differ in their exact epitope, yet likely target a proximal region within Domain 2 of gH (Fig. 8b). Collectively, these data confirm the existence of an 'antigenic hotspot' within the central alpha helix region of gH capable of eliciting mAbs with potent and broad-spectrum neutralizing capacity.

**Humanized mAb 5C3 maintains specificity and potency.** To demonstrate the therapeutic potential of murine neutralizing CMV mAbs, the sequence of mAb 5C3 was engineered to create a chimeric mAb consisting of the murine Fv domains fused to human constant domains, as well as a humanized antibody. Flow cytometry staining indicated that the re-engineered mAbs bind U373$^{gH/gL}$ cells to the same extent as murine 5C3 (Fig. 8i). AD169$_{IE2-YFP}$-neutralization assays conducted in MRC5 fibroblasts demonstrated nearly identical IC50 values for all three of the mAb variants (Fig. 8j). Together these data demonstrate that chimeric and humanized variants of mAb 5C3 do not display loss in specificity or neutralizing potency and exhibit the feasibility of

functional murine hybridoma screening for the development of novel therapeutic mAbs.

## Discussion

We utilized a high-throughput infection platform based on fluorescent reporter viruses to identify broadly neutralizing CMV mAbs targeting the gH envelope protein. The neutralizing mAbs were effective at blocking diverse strains of virus and different cell types by binding to the trimer and PCs (Fig. 5 and Supplementary Table 3). Thus, targeting a functionally relevant and conserved region of the gH protein is an attractive strategy for generating broad-spectrum biologics.

CMV-gH likely contributes to viral entry primarily through activation of the fusion event, rather than serving a receptor-binding role[48]. Given that 10C10 and 5C3 block infection at a post-attachment step (Fig. 6), the mAbs may function by interrupting a fusion-triggering signal to gB following viral attachment. Alternatively, gH may constitute part of the fusion complex and thus the mAbs directly interfere with the fusion event. Interestingly, the α-gB mAbs (Fig. 3) neutralized only AD169 and not the TB40/E strain; despite the mAbs' ability to bind to gB derived from TB40/E (Fig. 3d). These results suggest that alternative viral envelope proteins in TB40/E may compensate for mAb-mediated inhibition of gB to allow for CMV entry. Together, these data indicate that assessing neutralization by laboratory-adapted viral strains is insufficient for determining the utility of a therapeutic mAb.

The coevolution of CMV with its human host over millions of years has enabled it to evade immune clearance[49,50]. Besides well-characterized strategies to evade T cell and NK cell activation, CMV may limit the generation of neutralizing antibodies to essential domains of its envelope[50]. Thus, generating antibodies in mice immunized with purified virions will allow for the identification of neutralizing mAbs targeting functionally critical regions of envelope proteins that are obscured in the human host. Supplementation of serum from pregnant women with mAb 5C3 demonstrated enhanced CMV-neutralizing ability regardless of serostatus (Supplementary Fig. 9). Neutralization of both fibroblast and epithelial cell infection by Cytogam was also

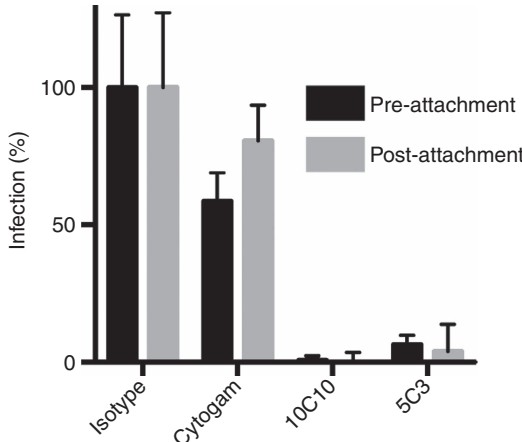

**Figure 6 | The anti-gH mAbs 10C10 and 5C3 neutralize infection at a post-attachment step.** AD169$_{IE2-YFP}$ was incubated with MRC5 cells at 4 °C followed by addition of Cytogam, an isotype control or anti-gH mAbs 10C10 and 5C3. Following an additional incubation at 4 °C, cell/virus/antibody were transferred to 37 °C. Infection levels (grey bars) were measured at 16 hpi and compared with virus prep that had been pre-incubated with mAbs before exposure to cells (black bars). Experiment was performed in technical triplicate and was replicated four times. S.d. is depicted for all experiments.

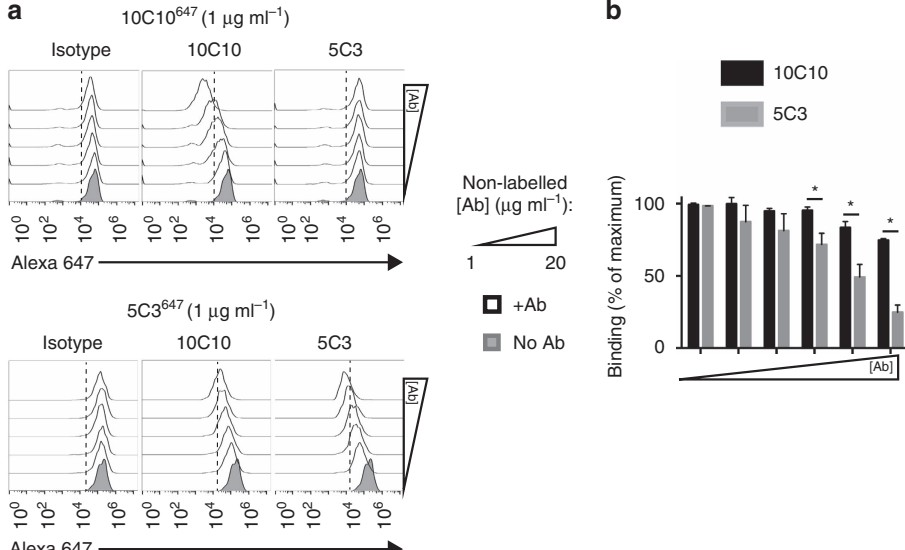

**Figure 7 | mAbs 10C10 and 5C3 bind to a similar region within gH.** (**a**) U373$^{gH/gL}$ cells were left unstained (grey peak) or were labelled with mAbs 10C10 (top row) and 5C3 (bottom row) conjugated to an Alexa647 fluorophore (10C10$^{647}$ and 5C3$^{647}$), together with increasing concentrations of non-conjugated 10C10 or 5C3 (1–20 μg ml$^{-1}$; white peaks). (**b**) The percentage of cells stained by 5C3$^{647}$ in the presence of non-labelled 10C10 (black bars) or non-labelled 5C3 (grey bars) is depicted for all concentrations. Experiment was performed in technical duplicate and was replicated four times. S.d. is depicted for all experiments. *$P < 0.05$ (Student's two-tailed $t$ test).

augmented by the addition of 5C3 (Supplementary Fig. 9) indicating that human antibodies to the 10C10/5C3 epitope region is minimally represented in human serum. This underscores the usefulness of murine-generated antibodies in developing antiviral biologics.

The 20aa region of gH Domain 2 surrounding the 10C10 and 5C3 epitope is 100% identical among geographically distinct CMV strains (Fig. 8h), despite sequence variation of gH within those strains[51,52]. Interestingly, mAb 14-4b binding to gH requires the wild-type residues 481–488 and 671–675 (ref. 47)

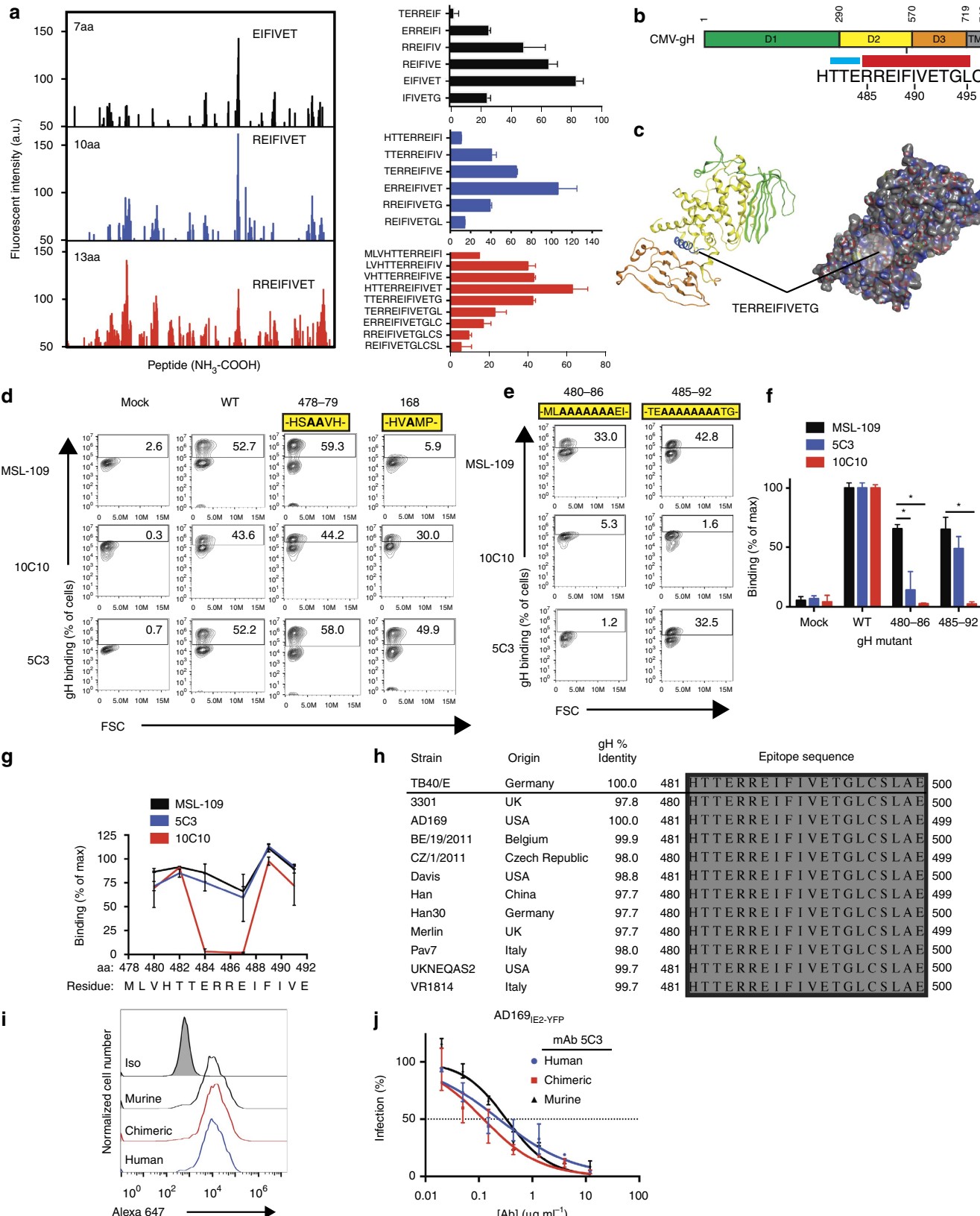

suggesting a potential overlap with the 5C3 and 10C10 epitope. Analysis of neutralization efficacy (IC50 values), epitope mapping using gH mutants, and competition studies (Supplementary Fig. 8) demonstrate that the mAbs' epitopes possess different binding requirements and do not overlap. Collectively, these findings support the existence of a highly immunogenic region of the gH protein susceptible to antibody-mediated neutralization using murine-derived mAbs.

Additionally, the MSL-109 epitope[53] has been identified to bind to Domain 2 of gH, yet MSL-109 binding to gH is dependent on essential residues in Domain 1. Generation of MSL-109 escape mutants identified residues W168, P171 or D446 as important residues for gH binding and neutralization[46]. Consistent with the model that 5C3 and 10C10 do not share an epitope with MSL-109, 5C3 and 10C10 strongly bind to W168A (Fig. 8d). Further, the epitope of MSL-109 may include multiple regions including aa 380–397, 403–423 and 442–446 (refs 45,53) that does not include the 5C3 and 10C10 epitope of residues 485–493. The inability of alanine pair substitutions across aa480–486 to abrogate 5C3 binding indicates that a significant sequence disruption within this region may be required to abrogate 5C3 binding. This implies that CMV may not easily escape 5C3 neutralization, as a considerable sequence alteration may be necessary in order for virus quasi-species to evade antibody-mediated inhibition. This may be an especially important characteristic for a neutralizing anti-gH mAb given that escape mutants to MSL-109 have been observed in vitro[46].

Advancements in antibody engineering permit the facile conversion of murine mAbs into human biologics[54]. In fact, chimeric and humanized variants of mAb 5C3 revealed no change in specificity or neutralizing potency of the mAbs (Fig. 8i,j). Further, the comparison of mAb 5C3 with the human-derived α-gH mAb, MSL-109 in neutralization assays in both fibroblasts and epithelial cells revealed nearly identical IC50 values (Supplementary Fig. 10) further demonstrating the value of murine-derived neutralizing mAbs. Human mAbs directed to gH or PC have been identified following a labour-intensive isolation and neutralization strategy[31], however these exhibit limited efficacy in neutralization of fibroblast cells. The α-gH mAbs identified to date bind to diverse regions of gH (Supplementary Fig. 11) and their mechanism(s) are not well-characterized[46,53]. Assessment of the clinical efficacy of MSL-109 did not demonstrate any benefit[55–58] possibly due to suboptimal dosing, viral escape or epitope variability among strains[46,59]. Also, its function may rely on disruption of gH/gL complexes that may be compensated for by alternative envelope complexes[46]. Thus, targeting a functional domain with an essential role in viral fusion would provide a more effective means of blocking infection.

Alternatively, a cocktail of α-gB, -gH and -PC mAbs may be the most effective strategy to block infection while preventing the generation of resistant strains[60–63].

In conclusion, we developed a high-throughput platform to identify a panel of potent CMV-neutralizing α-gH mAbs that recognize a highly conserved domain exposed in both the gH/gL/gO complex and the PC. These novel mAbs can be advanced for use in immunotherapeutics, in delineating CMV entry pathways, and for identification of critical protein domains to be targeted by vaccination. Moreover, the platform described here represents a rapid and effective framework for creating potent and broadly specific CMV biologics. Identification of the 10C10 and 5C3 epitopes points to the existence of conserved and highly immunogenic regions of the CMV-gH protein and provides valuable insight for the development of future antiviral immunotherapeutics.

## Methods

**Cell lines, antibodies and viruses.** Human U373-MG astrocytoma cells (gift from Dr Hidde Ploegh, Whitehead Institute, MIT) and MRC5 lung fibroblasts (ATCC #CCL-171) were cultured in Dulbecco's modified Eagle's medium (DMEM) supplemented with 10% foetal bovine serum (FBS). THP-1 cells (ATCC #TIB0202) were cultured in Roswell Park Memorial Institute (RPMI) supplemented with 10% FBS. Gp2–293 cells (Takara #631505) were employed to generate retroviruses and were cultured in identical conditions. Human retinal epithelial ARPE-19 cells (ATCC #CRL-2302) were cultured in DMEM and F-12 (1:1) with 10% FBS. All cell lines were tested for mycoplasma contamination. Culture media was supplemented with 1 mM HEPES, 100 U ml$^{-1}$ penicillin, and 100 g ml$^{-1}$ streptomycin at 37 °C in a humidified atmosphere (5% $CO_2$). The monoclonal anti-gH antibodies (clone 14-4b (ref. 64) and AP86 (ref. 65)), and the anti-gB antibody 27–156, kind gifts from William J. Britt, were purified from hybridoma culture supernatant. Polyclonal anti-gL immunoglobulins were raised in rabbits by inoculation with a peptide derived from the CMV TB40/E gL sequence (aa. 265–278, PAHSRYGPQAVDAR). Monoclonal antibody W6/32 (anti-MHCI; gift from Dr Hidde Ploegh, Whitehead Institute, MIT), which recognizes properly folded MHC class I molecules was purified from hybridoma culture supernatant. Anti-glyceraldehyde-3-phosphate dehydrogenase (GAPDH) was purchased (Chemicon, Billeriza, MA). CMV virus was propagated in MRC5 cells and virus from infected-cell supernatant and cell lysate (pooled) following sonication was purified by density gradient centrifugation by spinning at 20,000 r.p.m. at room temperature for 1.5 h over a 20% sorbitol cushion[35].

Infectious virus yield of AD169$_{IE2-YFP}$ (a gift from Dr Leor Weinberger (Gladstone Institute UCSF) was assayed on MRC5 fibroblasts by median tissue culture infectious dose (TCID$_{50}$). All other strains were titered by plaque assay.

**Production of murine hybridomas.** Five mice were immunized with AD169 (100 µg) followed by boost of 50 µg of virus preparations. Following the analysis of CMV neutralization using sera from the mice, two of the mice with highest neutralization titre were euthanized and their spleens were utilized to generate B-cell hybridomas using a standard protocol. The individual B-cell clones were selected from soft agar with a robotic Hamilton/Stem Cell Technology ClonaCell Easy Pick instrument, allowing direct screening of individual clones to identify those that neutralize a CMV infection.

**Figure 8 | Characterization of the α-gH mAb epitopes.** (**a**) Spot intensities were quantified following incubation of 5C3 followed by anti-mouse Dylight680 mAb with overlapping conformational gH peptide libraries of 7 amino acids (aa; black), 10aa (blue) and 13aa (red; left panel). Reactivity of 5C3 with overlapping peptide series near the region of peak binding are shown (right panel). Data represents mean intensity from duplicate samples. (**b**) The region of 5C3 reactivity is located between aa's 481–495 within domain 2 of gH. The region includes a β-sheet (blue) and a 10aa-long alphahelical region (red). (**c**) Structural modeling of CMV-gH based on the crystal structure of HSV-1 indicates the location of the putative epitope (blue; left). Predicted surface interactions indicate that the 5C3-reactive alpha helical region is exposed on the surface of gH (circle; right). (**d**) 293 cells transfected with gL and a transfection control, wild-type gH, gH mutant 478–79 or gH mutant 168 were stained with MSL-109 (top row), 10C10 (middle row) or 5C3 (bottom row). Percentage of cells positive for gH are indicated. (**e**) 293 cells transfected with gL and gH mutant 480–86 or 485–92 were stained with MSL-109 (top row), 10C10 (middle row) or 5C3 (bottom row). Percentage of cells positive for gH are indicated. (**f**) The data from Fig. 6e was quantified and normalized compared with WT binding. (**g**) Binding of MSL-109 (black), 5C3 (blue) and 10C10 (red) to 293 cells transfected with gH containing 2aa alanine substitutions along the length of the epitope region was measured. Percentage of gH-positive cells compared with WT-transfected cells was calculated and plotted. (**h**) The epitope region from 12 geographically distinct CMV strains were aligned to the TB40/E sequence. (**i**) U373$^{gH/gL}$ cells were exposed to murine, chimeric and humanized 5C3 mAbs and analyzed by flow cytometry with cells exposed to an isotype control (grey peak). (**j**) Murine, chimeric and humanized 5C3 mAbs were pre-incubated with AD169$_{IE2-YFP}$ at various concentrations before exposure to MRC5 fibroblasts. Infection levels were measured and compared with an isotype control. Data in **d**–**g**, **i**, **j** represents duplicates from three experiments. S.d. is depicted in all relevant figures. *$P < 0.05$ (Student's two-tailed $t$ test).

**High-throughput neutralization assay.** AD169$_{IE2-YFP}$ was pre-incubated with hybridoma supernatant (25 μl of supernatant, 25 μl of DMEM for total inoculum volume of 50 μl; MOI: 3) and incubated for 2 h at 4 °C. Development of the neutralization assay by varying incubation time and temperature did not limit virus infection on the incubation of virus at 4 °C (Supplementary Fig. 12A). The incubation of virus with supernatant for 2 h at 4 °C was selected due to the logistics of analysing a large number of hybridoma clones. The inoculum was then added to MRC5 cells (10,000 cells per well) in a 96-well plate. After 2 h the virus/antibody inoculum was removed and replaced with 100 μl DMEM. The plate was then read in an Acumen $^eX3$ laser-scanning fluorescence microplate cytometer to measure YFP fluorescence levels at 16 hpi. Serum samples were diluted with DMEM before virus incubation and then pre-incubated with AD169$_{IE2-YFP}$ as described above. Further, neutralization assays in which AD169$_{IE2-YFP}$ was pre-incubated with 5C3 for different times and temperatures demonstrated equivalent inhibition of infection (Supplementary Fig. 12) demonstrating the versatility of HTN assay. Experiments involving TB40/E$_{Flag-YFP}$ (TB40/E-Bac clone (gift from Dr Christian Sinzger (University of Ulm, Germany) was used to generate TB40/E$_{Flag-YFP}$) were conducted as described above and analyzed by fluorescent cytometer at 72 hpi. Following initial screen, all samples were tested in technical replicates of 3 to avoid confounding data from outliers.

**Immunostain assay.** Infected cells were fixed with 4% paraformaldehyde at 16 hpi for 15 min. at 4 °C. Cells were permeablized with 0.1% Triton X-100 and stained with mAb 810-X (1:250) (Millipore), before analysis by fluorescent cytometer.

**Plaque reduction assay.** MRC5 cells were seeded in duplicate with DMEM at a density of $5 \times 10^4$ cells per well in a 24-well plate (BD-Falcon, Franklin Lakes, NJ). The next day cells were infected with virus that was pre-incubated with a range of mAb concentrations (MOI 0.002). Following infection (2 h at 37 °C), cells were washed 2x with DMEM and wells were refilled with 3% DMEM containing DMSO supplemented with 10 μg ml$^{-1}$ Cytogam (CSL Behring, King of Prussia, PA). At 7–10 dpi, cells were fixed in 4% paraformaldehyde (20 min at 4 °C), and stained with Giemsa (Harleco, EMD Millipore; 1 h at 37 °C). Wells were washed with dH$_2$O $5 \times$ and plaques were blind counted using phase contrast microscopy.

**THP-1 neutralization assay.** THP-1 cells (250,000 cells per well) were infected with TB40/E$_{FLAG-YFP}$ that was pre-incubated with 0.5 μg ml$^{-1}$ mAb (MOI 3). Following infection (37 °C, 2 h) inoculum was removed and cells were washed 1 × with RPMI. Cells were cultured for 72 h and then analyzed for YFP fluorescence by flow cytometry on an Intellicyt HTFC flow cytometer.

**Transwell assays.** MRC5 cells (10,000 per well) were seeded on permeable inserts. The next day cells were infected with TB40/E (MOI: 1). Following inoculation (2 h at 37 °C), cells were washed 2 × and wells were filled with fresh media. At 2 dpi the well insert was transferred to a receiver plate containing either MRC5 cells or ARPE-19 cells (10,000 cells per well) that were seeded the previous day. Wells were then filled with media supplemented with various concentrations of mAb, such that both layers of cells were exposed to the media, and the permeable insert layer did not directly contact the bottom receiver layer. At 7 dpi receiver layer cells were fixed and analyzed by immunostain.

**ARPE-19 cell-cell dissemination assay.** ARPE-19 cells were seeded in a 96-well plate (10,000 cells per well) and the following day infected with TB40/E (MOI: 0.1). At 48 hpi, cell supernatant was replaced with media containing various concentrations of α-gH mAb, Cytogam or an isotype control. At 10 dpi, cells were fixed and analyzed by immunostain.

**Monocyte-fibroblast coculture experiments.** Monocytes were infected with TB40/E (MOI: 3) in RPMI with 1% FBS for 1 h at 37 °C with shaking every 15 min and then washed twice with 1 × PBS before culturing in complete RPMI containing 500 U ml$^{-1}$ granulocyte–macrophage colony-stimulating factor. Following three days of subculture, TB40/E-infected monocytes were transferred to fibroblasts monolayers containing equal number of cells and a range of mAb concentrations. At 6 dpi, monolayers were analyzed by immunostain and quantified using a fluorescent cytometer.

**Generation of stable U373 cell lines.** gH, gL, gO and UL128 cDNA from the CMV TB40/E sequence was cloned into the pLGPW, pLNCX and pLHCX vectors (Clontech) and stably introduced into U373 astrocytoma cells by retroviral transduction. The gO nucleotide sequence was optimized for expression in human cells.

**Flow cytometry analysis.** Flow cytometry analysis was performed as previously described[66]. Briefly, cells were incubated with 1%BSA/PBS, and then antibody (2 μg ml$^{-1}$) followed by secondary staining with Alexa-647 goat anti-mouse antibody (1:500) (ThermoFisher Scientific). Fluorescent data was collected on an

Intellicyt HTFC flow cytometer. For analysis of permeabilized cells, cells were fixed with BD Biosciences Cytofix/Cytoperm solution (20 min. at 4 °C) and maintained in 0.1% saponin throughout the staining procedure. The data were quantified using Flow Jo software (Tree Star, Inc).

**Cell lysis and immunoprecipitation.** Total cell lysates from U373 transgenic cell lines or infected MRC5 cells were subjected to SDS–PAGE followed by immuno-blot analysis using the anti-gH mAb AP86 (1:5 hybridoma supernatant), polyclonal anti-gL IgG (1:1,000), and an anti-GAPDH (1:10,000; Chemicon, Billeriza, MA). Uncropped immunoblot and autoradiograph films are displayed in Supplementary Fig. 13.

**Metabolic labelling analysis.** MRC5 fibroblasts were infected and pulsed at 72 hpi with $^{35}$S-methionine for 6 h at 37 °C. Cells were lysed in NP40 and immunoprecipitated using neutralizing gB mAbs 2F4, 5A6, 7H7, 8H2 or gH mAbs 10C10 and 5C3. Anti-GFP antibody was used as an isotype control. Following incubation of immune complexes with protein A agarose beads (RepliGen), samples were treated with 1 × SDS sample buffer and resolved using SDS–PAGE. For experiments involving N-glycanase F treatment, following immunoprecipitation the protein A beads were split into 2 fractions and one fraction was incubated with 500 units of PNGase F (New England Biolabs) for 2 h before addition of sample buffer and resolution by SDS–PAGE.

**Pre/post-attachment neutralization assay.** MRC5 cells (10,000 cells per well) were seeded overnight and the following day incubated at 4 °C, 30 min. AD169$_{IE2-YFP}$ virus prep (MOI: 3) was chilled to 4 °C and exposed to the cells for 30 min. followed by addition of pre-chilled mAbs (50 μg ml$^{-1}$ final concentration) and a final 30 min. incubation step at 4 °C. For pre-incubation with mAbs, virus was pre-incubated with mab at 4 °C, 2 h before exposure to cells. Cells were transferred to 37 °C for 2 h, followed by a 2 × DMEM wash. Fluorescence levels were measured at 16 hpi by fluorescent cytometer.

**Analysis of truncation mutants.** gH mutants were cloned into pCDNA 3.1 and transfected with lipofectamine. Cells were harvested 2 days post transfection and analyzed by Immunoprecipitation followed by resolution with SDS–PAGE and immunoblot with a monoclonal anti-HA antibody (12CA5; 1 μg ml$^{-1}$).

**Virion binding assay.** TB40/E cell-free virus prep was blocked with normal rabbit serum at 1:500, (4 °C, 1 h.). A range of mAb concentrations was then added and incubated at 4 °C, 2 h. Virus/antibody mixture was then ultracentrifuged at 24,000 r.p.m. in a SW55ti rotor for 90 min over a 20% sorbitol cushion. Supernatant was removed and the pelleted virus/antibody was resuspended with 1 × sample buffer and resolved by SDS–PAGE followed by detection with an anti-mouse HRP antibody. Immunoblot using Cytogam permitted the visualization of virus levels.

**Mass spectrometry proteomic analysis.** mAb 10C10 (10 μg) was incubated with lysates from TB40/E-infected (MOI: 3) MRC5 cells (10 million) at 72 hpi. The precipitates were directly subjected to quantitative analysis of the protein contents at Bioproximity, Inc (Chantilly, VA) using global proteome profiling by LC-MS/MS utilizing Orbitrap mass spectrometry. The protein intensity was calculated as the sum of the peptide-associated features that map to a protein sequence. Protein intensities were adjusted (1) by normalization based on total intensity within a LC-MS/MS run and then across all considered runs, and (2) by background subtraction. Multiple quantitation methods were supported as MS1 peak area, MS2 intensity and spectral counting-derivatives. Proteins were required to have one or more unique peptides across the analyzed samples with low E-value scores. The mascot generic format files were searched using the most recent human and CMV sequence libraries obtained from UniProt. All searches were performed on Amazon Web Services-based cluster compute instances using the Proteome Cluster interface.

**Generation of gH peptide library.** The C- and N-terminus of gH was elongated by neutral GSGSGSG linkers to avoid truncated peptides. The elongated sequence was translated into 7, 10 and 13 amino acid peptides with peptide-peptide overlaps of 6, 9 and 12 amino acids. After peptide synthesis, all peptides were cyclized via a thioether linkage between a C-terminal cysteine side chain thiol group and an appropriately modified N-terminus. The resulting cyclic peptide microarrays contained 1,815 different cyclic peptides printed in duplicate (3,630 peptides spots) and were complemented on right and left of the array by additional HA (YPYDVPDYAG) and Flag (DYKDDDDKGG) control peptides (60 spots each control).

**Analysis of epitope reactivity of the gH microarray.** Incubation of peptide microarray libraries was carried out with mAb 5C3 at concentrations of 10 μg ml$^{-1}$, 100 μg ml$^{-1}$ and 500 μg ml$^{-1}$ in incubation buffer followed by

staining with the secondary goat anti-mouse IgG (H + L) DyLight680 antibody and read-out at a scanning intensity of 7 (red). Additional HA peptides on right and left of the peptide arrays were finally stained as internal quality control to confirm the assay quality and the peptide microarray integrity (scanning intensities: 7/7, red/green). Quantification of spot intensities and peptide annotation were based on the 16-bit grey scale tiff files at scanning intensities of 7 that exhibit a higher dynamic range than the 24-bit colorized tiff files; microarray image analysis was done with PepSlide Analyzer. An intensity map based on averaged foreground intensities was generated.

**Serum supplementation experiments.** Human serum (1:1,000) was supplemented with a range of mAb concentration and pre-incubated with TB40/E (MOI: 3; 4 °C, 2 h). Following incubation, virus/serum/mAb mixtures were added to MRC5 or ARPE-19 cells (10,000 cells per well). Following infection (37 °C, 2 h), inoculum was removed and cells were washed 2 ×. Infection levels were analyzed at 16 hpi by anti-IE immunostain and fluorescent cytometer.

**mAb binding competition experiment.** mAbs 10C10 and 5C3 were conjugated to Alexa Fluor 647 using a labelling kit (ThermoFisher). U373$^{gH/gL}$ cells were blocked with 1% BSA/PBS and then simultaneously exposed to a fixed concentration of Alexa647-conjugated antibody, and a range of concentrations of non-labelled antibody. Fluorescent data was collected on an Intellicyt HTFC flow cytometer. The data were quantified by using Flow Jo software.

**Chimerization/humanization of 5C3.** Sequencing and chimerization for 5C3 variable heavy and kappa chains were obtained by an adaptation of methods previously published for single mouse cells[67]. Briefly, RNA was extracted from the 5C3 hybridoma using Qiagen RNeasy Mini Kit (Qiagen, Valencia, CA), followed by first stand cDNA synthesis using random hexamers and poly-T primers (Integrated DNA Technologies, Caralville, IA) and SuperScript 3 reverse transcriptase (ThermoFisher Scientific, NY, NY). PCR with FastStart Taq DNA polymerase (Roche) was performed according to factory supplier with degenerate broad primers specific for 5′ UTR ends of heavy and kappa mouse V genes and specific IgG constant and kappa constant 3′ primers[67]. Purified PCR product was then submitted for Sanger sequencing in both directions using same primers as PCR (GeneWiz, South Plainfield, NJ). Sequence results were blasted against the IMGT mouse databank of germ-line genes using V-Quest (http://imgt.org) and specific 5′ and 3′ primers were built to the exact 5′ start of the V gene and the exact 3′ end of the J gene. For 5C3, sequence blasted as a mouse VH1–61/JH3 for heavy and mouse VK 3–12/JK2 for kappa. Cloning PCRs using specific cloning primers based on germ-line gene information were performed from cDNA and cloned into pFuseSS human IgG1 and human kappa constant containing vectors (InvivoGen, San Diego, CA). Clones were transfected into human Expi293 cells and chimerized antibody was purified from supernatant using protein A columns on an Akta FPLC protein purification system (GE Healthcare Life Sciences, Pittsburgh, PA).

Mouse 5C3 (parental) was humanized by germ-line framework shuffling following methodologies in ref. 68 with each mouse framework region (FW1–4) for the heavy and kappa chain independently replaced with a human acceptor framework region of highest homology. In brief, the process of humanizing started with prediction of 5C3 Fv structure using the ROSIE online server for antibody structure (http://rosie.rosettacommons.org/antibody)[69,70]. Using a Kabat numbering scheme, framework residues for heavy and light chain were blasted independently using NCBI IgBlast (http://www.ncbi.nlm.nih.gov/igblast/). When disagreements between parental and human residues were encountered, residues were changed to human unless the residue was identified as a key residue or in a structurally relevant position. In this case, residues were evaluated individually and in context of structure model, and parental sequence was maintained (backmutation) if a residue was hypothesized to greatly compromise structural integrity and/or binding activity of antibody. The human acceptor framework sequences containing all identified backmutations for each framework were then grafted between mouse CDR sequences in silico using Geneious DNA software (Biomatters Limited, Newark, NJ) and were synthesized as gBlocks (Integrated DNA Technologies) containing same 5′/3′ restriction sites as for chimerization. Constructs were cloned into same human constant gene containing vectors, transfected and purified from supernatant as for chimerization.

**Statistical analysis.** Student's unpaired, two-tailed $t$ tests were performed using GraphPad Prism (La Jolla, CA). Asterisks represent statistical significance of $P < 0.05$ or less. For calculation of half maximal inhibitory concentration (IC50), 4 parameter non-linear regression analysis was performed using GraphPad Prism. Standard deviation is depicted in all relevant figures.

**Data availability.** All data are available from the authors upon request.

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

## Acknowledgements

This work was supported in part by the NIH grants AI101820 and AI112318. T.J.G. was a pre-doctoral trainee supported in part by an American Heart Association pre-doctoral fellowship. T.J.G. and T.M.S. are in part supported by the pre-and post-doctoral USPHS Institutional Research Training Award T32-AI07647.

## Author contributions

T.J.G. contributed significantly to the acquisition, analysis and interpretation of the data, study design and drafting of the manuscript. K.R.S., J.A.D., T.M.S., V.M.N. and T.K. contributed significantly to the acquisition of data and critical review of the manuscript. T.M.M. and D.T. contributed significantly to the interpretation of data, study design, drafting and critical review of the manuscript.

## Additional information

**Competing financial interests:** The authors declare no competing financial interests.

