## [Peer Review File · Nature Communications]

Reviewers' comments:

Reviewer #1 (Remarks to the Author):

This is an exciting paper from Tortella and colleagues, who have developed a HTP strategy for the isolation and characterization of neutralizing antibodies to HCMV. These studies have identified novel mAbs that target the gH glycoprotein and specifically gH as part of the trimer (gH/gL/gO) or pentamer (gH/gL/UL128-UL131) complex. This process has identified novel "antigenic" hot spots critical for virus entry. The results not only provide a solid foundation for the development of potent therapeutic antibodies against HCMV but also have important implications for the development of an effective HCMV vaccine strategy. This is a very well written paper with in depth experiments presented as main figures and supplementary data. The figures are well laid out and provide a basis for reaching a series of clear conclusions. The data and methodology are sound and results extensive, reliable and well characterized. The abstract accurately reflects the paper.

A minor criticism is that the studies only investigate virus infection of one epithelial cell type and additionally infection/inhibition of endothelial cells is not evaluated. However, potentially the scope of the paper limits these options.

Although the concept and generation of neutralizing antibodies against HCMV for use as therapeutics is not novel in the CMV field. The approach and findings of the paper are relatively exciting and novel. The paper has the potential to have a high impact in CMV intervention research as well as enable further evaluation of key aspects of the process of CMV cell entry.

I am highly in favor of the manuscript being accepted for publication without further modification.

Minor typographical error L183: "gH/gL/128" should be "gH/gL/UL128"

Reviewer #2 (Remarks to the Author):

Overall, the approaches used by Gardner et al are sound, and the data, with some notable exceptions, support the conclusions. Enthusiasm for this manuscript is limited because, as detailed in point 13 below, the anti gH antibodies described do not appear to be "novel", as suggested by the manuscript title. Rather, they seem to be more or less similar (both in epitope specificity and neutralizing characteristics) as the previously studied mAbs 14-4b and MSL109. A number of concerns and suggestions are offered for the authors' consideration below.

1. Lines 34-38: 1) There are no data in Compton 1992 suggesting that gH-trimer is required or, even involved in fibroblast entry. Better refs would be Willie 2010 JVI, Jiang 2008 JVI, and Zhou 2015 JVI. 2) Ryckman 2006 shows that agents that inhibit endosome acidification

block HCMV infection of epithelial and endothelial cells, but (despite the paper title) does not show that the fusion event is, in fact "pH-dependent". (and FYI this ref was left out of the citation string). 3) as written these two sentences seem to perpetuate the notion that gH-trimer and gH-pentamer are sufficient for entry into fibs and epi/endo, respectively. Consider revising to phrase more like in the abstract lines 13-16, with more appropriate refs.

2. Line 62-16; fig 1a. How can "mock-infected cells" be "normalized as 100% infection?" This is confusing. Do the authors mean "mock-neutralized virus"? As in, HCMV not treated with any potentially neutralizing antibody? This would seem to make more sense.

3. Fig 1. Either the labels on the plots themselves, or the legend should explain which antibodies were used in each experiment, so that the reader does not need to refer to the results section text. For example, it seems that mAb 14-4b is labeled as "a gH" in panels A and D. Is this correct?

4. Please indicate whether HCMV used in experiments was "cell-free" derived from culture supernatants, or derived from sonicated cells, or other.

5. Lines 338-346; "Fluorescent virus neutralization assay" 1) This seems to be the same thing as "high-throughput neutralization; HTN"? If so, consider being more consistent in terminology throughout the manuscript. 2) It seems that virus and antibody were incubated at 4 degrees for 2h prior to being added to cells. Assuming the "100%-infection" normalization was, as the reviewer suggested in point 2 above, a "no-antibody control", were these also incubated for 2 hr at 4 degrees? Either way, this is potentially a problematic protocol since HCMV is highly labile at 4 degrees. This phenomenon was described many decades ago (see Plumber and Lewis, J. Bact. 1965; Vonka and Benyesh-Melnik, J. Bact, 1966) and has also been observed/verified by many "contemporary HCMV labs", including mine. As much as 10-fold loss in titer can occur very rapidly. The "mechanism" of this inactivation is not clear, but the virions are still in the sample. It is hard to predict how this might affect the outcome of antibody neutralization reactions. If nothing else, an assay performed this way is measuring the sensitivity of a sub-population (the most stable at 4 degrees) of HCMV to the antibody in question. Furthermore, given that the major aim of this research is towards development of a vaccine that will elicit antibodies, which must work at 37 degrees/body-temp, this might be a problem.

6. Fig 2 A and B. IP from infected cells. These data are broadly consistent with reactivity to gB-species for the "group I" and for gH-species for "group II." However there are some peculiarities with the group II Abs that could be mentioned/explained. 1) 5C3 seems to pull down less gL and UL128 than 10C10, but it is hard to tell if this reflects the differences in gH pull down. Can band density ratio be quantified for comparison? E.g., gH:gL and gH:UL128. 2) If 5C3 and 10C10 are pulling down gH, gL and UL128, some readers might also expect to see some UL130 and UL131. Maybe point out that the AD169 used does not express a full length UL131. However, still it seems there should be some UL130, no? 3) The PNGaseF band shifts all look good. Except that the authors could mark UL128 on the gel, and note that it does not shift because it does not have any N-linked glycan sites.

7. Fig 3. So, the gB antibodies react with TB40 gB, but do not neutralize? This is an interesting result, but is not developed.

8. Fig 4A. The use of an anti-IE mAb for this HTN is conceptually reasonable. However, there are amino acid polymorphisms in IE proteins between these strains. Is the epitope for this mAb known? I assume that the neutralization curve for each strain is set at 100% for the no-Ab condition, but any differences in anti-IE reactivity between strains might mean that the "dynamic range" is different...thus complicating the comparison of IC50 between strains.

9. Lines 131-132. What is meant here? If an Ab blocks infection, then it must block "subsequent viral protein expression", no? This analysis is confusing, and seems to add little. Consider removing it to keep the reader on track with the main points.

10. Fig 4B. and lines 137-138. A plaque reduction experiment is quite reasonable, but as performed (according to M&M lines 352-359) it seems to show nothing beyond what was shown in Fig 4A (IE analysis); i.e., a block to the initiation of infection. Since you pre-treat the virus with Ab, then count the number of plaques that form, how does this say anything about "dissemination"? Simply seems to be an alternate readout to the experiment in 4A.

11. Fig 4E/F/G Lines 153-.168; Why do the authors' use the term "cell-to-cell"? It seems that to most virologists, this means direct spread of virus from one cell to another through sites of cell-cell contact, as distinct from spread of cell free virus through the extracellular environment (i.e., culture supernatant). The experiments presented here do not seem to distinguish cell-free spread from cell-to-cell spread. In fact, the transwell assay seems to eliminate the possibility of "cell-to-cell" spread because it physically separates the infected and uninfected cells. The only way for the virus to spread in this assay is via cell free virions released from the upper to the lower chamber. Thus, this experiment does not really say anything more than those in the previous experiments..i.e., these antibodies neutralize cell free virus.

The experiment in Fig4G is confusing since lines 161-163 indicate that ARPE19 cells were infected at a low MOI first, then antibody was added, and the readout was the number of plaques. Thus, the Y-axis on 4G, "% infection" does not seem appropriate. "Number of plaques (as % of control)" would seem to be better. But again, this experiment does not seem to be reflective of "cell-to-cell" spread but rather, spread of cell-free progeny through the culture supernatant to initiate new plaques. A cell-to-cell spread measure would need to include the size of each plaque. Even then, since HCMV strains like TB40/e can spread either cell-to-cell or cell free, a size reduction does not necessarily distinguish the modes of spread. In sum, the experiments in Fig 4 are all consistent with each other, and only one major conclusion seems evident, these antibodies neutralize cell-free virus of different strains of HCMV on both MRC-5 and ARPE19 cells.

12. Fig 5. It seems that the groups of 5A-C, 5D-E, and 5F are three independent lines of experiments, with independent questions, that would be better presented as 3 separate figures. Also, the text associated with D, E is very hard to follow (lines 193-202). It seems

that reference is often made to the wrong panel.

13. Fig 6 epitope mapping. The suggested epitope for 5C3 and 10C10 at 485-493 of gH exactly overlap with the epitope of 14-4b (and likely MSL-109) as suggested in Schultz 2016 JVI. This should be noted. And given that both 5C3 and 10C10 seem to have no new properties compared to those already described for 14-4b, how are these "novel"? The authors are encouraged to provide direct comparisons of their mAbs with 14-4b...esp in the competitive binding assays.

Reviewer #3 (Remarks to the Author):

In this interesting manuscript by Gardner and colleagues, there are significant and novel observations made about innovative methodologies to interrogate the humoral immune response to human cytomegalovirus infection. The over-arching theme is a focus on the so-called pentameric complex (PC) encoded by CMV. The key importance of the PC is that it mediates uptake of virus and subsequent infection in all cell types, including fibroblasts, epithelial cells, and myeloid cells. Hence, antibody to the PC is predicted to be broadly neutralizing to for all cell types, making it an ideal vaccine candidate for both active immunization and passive (antibody-mediated) immunization, such as might be administered to high-risk pregnant women with primary or recurrent CMV infection.

In this study, a novel robotics approach was taken, in combination with a high through-put neutralization assay, to screen for and subsequently identify a series of monoclonal antibodies target the CMV gH protein (an integral component of the pentameric complex that mediates cell entry). Detailed conformational analyses identified that the monoclonals identified bind to either the gH trimer formed with gH/gL/gO, or the gH pentamer formed by gH/gL/UL128/UL130/UL131. Key epitopes essential for viral entry are elucidated, and mechanisms of neutralization are described. One significant strength of the manuscript is that it goes beyond the descriptive nature of many similar studies to employ an informatics approach to rigorously define the relevant targets, including an alpha-helix rich domain that is targeted for neutralization.

Another strength of the manuscript is that the methodology described in the paper provides a framework for future discovery and design of vaccines and immune based therapies that will advance not only the CMV field, but antiviral immunologics for other pathogens. Hence, the manuscript will have broad interest to the readership, not only basic scientists and immunologists, but also vaccine designers, pediatricians and infectious diseases specialists, public health and policy experts, and anyone interested in childhood disabilities.

The manuscript in general is well-written and the experiments logically described. The conclusions are supported by the data and the controls are appropriate. A few areas for improvement are noted.

1. Lines 41-46. The authors should note that CMV immune globulin has not been definitively shown to protect the CMV-vulnerable fetus. Indeed, a controlled trial demonstrated no benefit (A randomized trial of hyperimmune globulin to prevent congenital cytomegalovirus. Revello MG, Lazzarotto T, Guerra B, Spinillo A, Ferrazzi E, Kustermann A, Guaschino S, Vergani P, Todros T, Frusca T, Arossa A, Furione M, Rognoni V, Rizzo N, Gabrielli L, Klersy C, Gerna G; CHIP Study Group. *N Engl J Med.* 2014;370(14):1316-26.). The authors should include this reference and may wish to add a comment on what the possible deficiencies were in the antibody preparation, and what improvements would be required (that the authors may be able to address) to improve the track record with immune globulin therapy in pregnancy.

2. In the screen, mice were immunized with AD169, precisely because it does not encode a functional pentameric complex (lines 68-70), and ~1500 candidate hybridomas were screened. A total of 6 highly neutralizing clones were subjected to further analysis. This demonstrated that mAbs 10C10 and 5C3 (group II) recovered proteins of ~100kDa, 37kDa, and 12kDa consistent with gH, gL and UL128 proteins. But it is unclear from the subsequent discussion (lines 90-105) whether the UL128 protein was recognized by this monoclonal. This point raised by the authors needs to be clarified. If this indeed was UL128, the authors should explain how this could be identified in AD169 virions, since UL128 is not supposed to be incorporated into virions in AD169 according to the work of other authors, since AD169 cannot generate a PC because of a mutation.

3. The gH monoclonals neutralized pentameric complex-containing clinical isolates more effectively than the gB monoclonals, compatible with a conformation-dependent targeting of gH in the pentamer. However, it is noted (line 129-129) that Cytogam was unable to block infectivity of these strains. This seems incompatible with the observations of Auerbach (reference 31) that the majority of the neutralizing activity in Cytogam targets the pentamer. The authors should reconcile this apparent discrepancy.

4. A strength of the study is the use of the novel PEPperCHIP® assay, consisting of 217 overlapping cyclic peptides corresponding to gH. Interestingly, the 5C3 monoclonal bound reproducibly to a peptide, RREIFIVET corresponding to domain 2 of gH. A 3D model of gH based on the structure of HSV-2 gH predicted that aa481-493 fall in an alpha helix domain on the surface of gH. The authors should comment, based on review of the literature, if this epitope has been reported by other investigators mapping anti-gH response. Is this a novel, heretofore unrecognized epitope? If so, this would add to the novelty and value of the manuscript.

5. The fact that the humanized 5C3 monoclonal continued to show specificity in flow cytometry staining and AD169 neutralization (lines 247-255). However it is not clear if neutralization assays with the humanized monoclonal were performed with the humanized variant. This would be predicted, but the authors should comment on this point since it greatly enhances the therapeutic potential of the antibody. This information might be in the figure legend for figure S7 but it is unclear. This point needs to be elucidated and clarified.

We thank the reviewers for their insightful comments to improve the manuscript! The italicized reviewers' comments were directly addressed by a response. We have also included all of the editorial changes recommended by the reviewers.

Reviewer 1 comments

Reviewer #1 (Remarks to the Author):

This is an exciting paper from Tortella and colleagues, who have developed a HTP strategy for the isolation and characterization of neutralizing antibodies to HCMV. These studies have identified novel mAbs that target the gH glycoprotein and specifically gH as part of the trimer (gH/gL/gO) or pentamer (gH/gL/UL128-UL131) complex. This process has identified novel "antigenic" hot spots critical for virus entry. The results not only provide a solid foundation for the development of potent therapeutic antibodies against HCMV but also have important implications for the development of an effective HCMV vaccine strategy. This is a very well written paper with in depth experiments presented as main figures and supplementary data. The figures are well laid out and provide a basis for reaching a series of clear conclusions. The data and methodology are sound and results extensive, reliable and well characterized. The abstract accurately reflects the paper.

A minor criticism is that the studies only investigate virus infection of one epithelial cell type and additionally infection/inhibition of endothelial cells is not evaluated. However, potentially the scope of the paper limits these options.

- Our studies demonstrate that the neutralizing mAbs inhibit CMV infection of fibroblasts and epithelial cells, two cell types with physiological importance. However, the reviewer is correct in that neutralization assays were not examined in endothelial cells, but was demonstrated in monocytes (Figure 4D), a cell type critical for virus dissemination and latency. Collectively, the manuscript demonstrates that the newly identified mAbs neutralize virus infection in three diverse cell types and are broadly neutralizing antibodies. This point is discussed throughout the manuscript.

Although the concept and generation of neutralizing antibodies against HCMV for use as therapeutics is not novel in the CMV field. The approach and findings of the paper are relatively exciting and novel. The paper has the potential to have a high impact in CMV intervention research as well as enable further evaluation of key aspects of the process of CMV cell entry.

I am highly in favor of the manuscript being accepted for publication without further modification.

Minor typographical error L183: "gH/gL/128" should be "gH/gL/UL128"

- This error was corrected.

Reviewer 2 comments

Overall, the approaches used by Gardner et al are sound, and the data, with some notable exceptions, support the conclusions. Enthusiasm for this manuscript is limited because, as detailed in point 13 below, the anti gH antibodies described do not appear to be "novel", as suggested by the manuscript title. Rather, they seem to be more or less similar (both in epitope specificity and neutralizing characteristics) as the previously studied mAbs 14-4b and MSL109. A number of concerns and suggestions are offered for the authors' consideration below.

- The revised manuscript has included numerous studies to demonstrate the novelty of the newly identified neutralizing monoclonal antibodies (Supplemental Figure 8) compared to other anti-gH antibodies that we discuss in detail below. In addition, the platform presented in the study using high-throughput assay and state-of-the-art technologies to identify and characterize broad spectrum neutralizing mAbs is quite novel and is a major advancement to the development of biologics to cytomegalovirus and other possibly pathogens.

2. *Lines 34-38: 1) There are no data in Compton 1992 suggesting that gH-trimer is required or, even involved in fibroblast entry. Better refs would be Willie 2010 JVI, Jiang 2008 JVI, and Zhou 2015 JVI. 2) Ryckman 2006 shows that agents that inhibit endosome acidification block HCMV infection of epithelial and endothelial cells, but (despite the paper title) does not show that the fusion event is, in fact "pH-dependent". (and FYI this ref was left out of the citation string). 3) as written these two sentences seem to perpetuate the notion that gH-trimer and gH-pentamer are sufficient for entry into fibs and epi/endo, respectively. Consider revising to phrase more like in the abstract lines 13-16, with more appropriate refs.*
 - We thank the reviewer for the suggestion and have included the recommended citations. We have reworded the section to more accurately convey the current state of knowledge regarding CMV entry.
3. *Line 62-16; fig 1a. How can "mock-infected cells" be "normalized as 100% infection?" This is confusing. Do the authors mean "mock-neutralized virus"? As in, HCMV not treated with any potentially neutralizing antibody? This would seem to make more sense.*
 - The reviewer is correct and we have made the suggested changes.
4. *Fig 1. Either the labels on the plots themselves, or the legend should explain which antibodies were used in each experiment, so that the reader does not need to refer to the results section text. For example, it seems that mAb 14-4b is labeled as "a gH" in panels A and D. Is this correct?*
 - We have updated the legend to include the reviewer's suggestions.
5. *Please indicate whether HCMV used in experiments was "cell-free" derived from culture supernatants, or derived from sonicated cells, or other.*
 - We have updated the materials and methods section to include this information.
6. *Lines 338-346; "Fluorescent virus neutralization assay" 1) This seems to be the same thing as "high-throughput neutralization; HTN"? If so, consider being more consistent in terminology throughout the manuscript.*
 - We have standardized the terminology in the revised manuscript.
7. *It seems that virus and antibody were incubated at 4 degrees for 2h prior to being added to cells. Assuming the "100%-infection" normalization was, as the reviewer suggested in point 2 above, a "no-antibody control", were these also incubated for 2 hr at 4 degrees? Either way, this is potentially a problematic protocol since HCMV is highly labile at 4 degrees. This phenomenon was described many decades ago (see Plumber and Lewis, J. Bact. 1965; Vonka and Benyesh-Melnik, J. Bact, 1966) and has also been observed/verified by many "contemporary HCMV labs", including mine. As much as 10-fold loss in titer can occur very rapidly. The "mechanism" of this inactivation is not clear, but the virions are still in the sample. It is hard to predict how this might affect the outcome of antibody neutralization reactions. If nothing else, an assay performed this way is measuring the sensitivity of a sub-population (the most stable at 4 degrees) of HCMV to the antibody in question. Furthermore, given that the major aim of this research is towards development of a vaccine that will elicit antibodies, which must work at 37 degrees/body-temp, this might be a problem.*
 - We greatly appreciate the reviewer's insight on this issue. In order to exclude this potential issue in the assay, we initially performed an experiment prior to the neutralization screen that varied the time of incubation and the temperature and found that that neither condition altered the number of infected cells based on fluorescence signal from IE2-YFP (Supplemental Figure 12). The rationale for selecting an incubation temperature at 4°C and 2hrs was due to the logistical considerations when managing a large number of multi-well plates during the initial hybridoma neutralization screen. In order to further demonstrate that the

incubation time and temperature did not affect antibody neutralization, a neutralization assay using 5C3 preincubated with virus at different times (1 and 2hrs) and temperature (4°C, 25°C, and 37°C) yielded similar inhibition of infection (Supplemental Figure 12). Thus, incubation of virus at 4°C does not have a negative impact on infection or neutralization.

- Additionally, the Plummer et al paper cited by the reviewer indicates that while CMV stability is negatively affected at 4°C, this instability does not significantly manifest prior to 6 hours of incubation (see figures 3 and 4 in Plummer et al.), whereas the pre-incubation in our assay was for only ~2 hours. The second paper by Vonka and Benyesh-Melnik demonstrates that cells incubated at 4°C during CMV infection develop fewer plaques – however our infections were all carried out at 37°C, thus our incubation/infection conditions should not contribute to decreased infection levels. Collectively, the conditions of virus/antibody incubation followed by infection outlined in the manuscript should not negatively impact virus infection.
 -
8. *Fig 2 A and B. IP from infected cells. These data are broadly consistent with reactivity to gB-species for the "group I" and for gH-species for "group II." However there are some peculiarities with the group II Abs that could be mentioned/explained. 1) 5C3 seems to pull down less gL and UL128 than 10C10, but it is hard to tell if this reflects the differences in gH pull down. Can band density ratio be quantified for comparison? E.g., gH:gL and gH:UL128. 2) If 5C3 and 10C10 are pulling down gH, gL and UL128, some readers might also expect to see some UL130 and UL131. Maybe point out that the AD169 used does not express a full length UL131. However, still it seems there should be some UL130, no? 3) The PNGaseF band shifts all look good. Except that the authors could mark UL128 on the gel, and note that it does not shift because it does not have any N-linked glycan sites.*
- We have included the recommended explanation of the absence of UL131 in AD169.
 - We were also initially surprised from these findings. However, these findings were consistent with published work (Wang and Shenk PNAS 2005) that an anti-gH immunoprecipitation from AD169-infected fibroblasts did not recover UL130. The Wang/Shenk manuscript also demonstrated that a repaired AD169 that expressed a full-length UL131 allowed for the recovery of UL130 from an anti-gH immunoprecipitate. In order to address this point, we have included recently acquired mass spec data that identified the components of the trimer and PC recovered from a 10C10 immunoprecipitation of TB40/E-infected cells (72hpi).
 - We have included a label of UL128 on the gel as recommended.
9. *Fig 3. So, the gB antibodies react with TB40 gB, but do not neutralize? This is an interesting result, but is not developed.*
- We agree that this is a very exciting result and we are currently investigating it! In the current manuscript we intended to focus on the development of a platform for mAb discovery as well as the identification of broad-spectrum neutralizing antibodies. We hope to publish the findings regarding the gB mAbs soon in a follow-up manuscript.
10. *Fig 4A. The use of an anti-IE mAb for this HTN is conceptually reasonable. However, there are amino acid polymorphisms in IE proteins between these strains. Is the epitope for this mAb known? I assume that the neutralization curve for each strain is set at 100% for the no-Ab condition, but any differences in anti-IE reactivity between strains might mean that the "dynamic range" is different...thus complicating the comparison of IC50 between strains.*
- The reviewer is correct that the dynamic range can vary slightly between viral strains using this assay. To our knowledge the mAb epitope is not known. While the IE1 sequence for VHL/E is not available, an NCBI BLAST alignment indicates that the IE1 protein sequences of TB40 and TR possess 99 and 98.6% similarity with that of AD169. Thus, we feel that any difference in IC50 values is unlikely due

to antibody recognition of the IE protein. However, expression levels and stability of IE1 as potential factors that determined the IC50 values. Hence, we focused our discussion on comparing the IC50 values between the different antibodies and Cytogam and less between the viruses.

- Additionally, in part due to the potential bias of the immunofluorescent assay, plaque assays were utilized as an alternative readout that would not be influenced by relative antibody affinity. We have updated the text to better rationalize the use of the plaque assay.

11. *Lines 131-132. What is meant here? If an Ab blocks infection, then it must block "subsequent viral protein expression", no? This analysis is confusing, and seems to add little. Consider removing it to keep the reader on track with the main points.*

- We agree and have deleted the analysis in the text.

12. *Fig 4B. and lines 137-138. A plaque reduction experiment is quite reasonable, but as performed (according to M&M lines 352-359) it seems to show nothing beyond what was shown in Fig 4A (IE analysis); i.e., a block to the initiation of infection. Since you pre-treat the virus with Ab, then count the number of plaques that form, how does this say anything about "dissemination"? Simply seems to be an alternate readout to the experiment in 4A.*

- The reviewer is correct in pointing out that the assay does not measure dissemination, and the text has been changed to reflect this. As addressed in comment 9, the plaque assay serves as an additional method to validate the efficacy of the anti-gH antibodies that is not biased by affinity of the antibody used for immunostain analysis.

13. *Fig 4E/F/G Lines 153-.168; Why do the authors' use the term "cell-to-cell"? It seems that to most virologists, this means direct spread of virus from one cell to another through sites of cell-cell contact, as distinct from spread of cell free virus through the extracellular environment (i.e., culture supernatant). The experiments presented here do not seem to distinguish cell-free spread from cell-to-cell spread. In fact, the transwell assay seems to eliminate the possibility of "cell-to-cell" spread because it physically separates the infected and uninfected cells. The only way for the virus to spread in this assay is via cell free virions released from the upper to the lower chamber. Thus, this experiment does not really say anything more than those in the previous experiments..i.e., these antibodies neutralize cell free virus.*

- The reviewer is correct and we have eliminated the use of the term "cell-to-cell" to describe this series of experiments. As we have now clarified in the materials and methods (see response 4), preceding neutralization experiments were performed with virus isolated from pooled supernatant and cell-associated virus. Thus the transwell experiment demonstrated the ability of the mAbs to block nascent cell-free virus.

14. *The experiment in Fig4G is confusing since lines 161-163 indicate that ARPE19 cells were infected at a low MOI first, then antibody was added, and the readout was the number of plaques. Thus, the Y-axis on 4G, "% infection" does not seem appropriate. "Number of plaques (as % of control)" would seem to be better. But again, this experiment does not seem to be reflective of "cell-to-cell" spread but rather, spread of cell-free progeny through the culture supernatant to initiate new plaques. A cell-to-cell spread measure would need to include the size of each plaque. Even then, since HCMV strains like TB40/e can spread either cell-to-cell or cell free, a size reduction does not necessarily distinguish the modes of spread. In sum, the experiments in Fig 4 are all consistent with each other, and only one major conclusion seems evident, these antibodies neutralize cell-free virus of different strains of HCMV on both MRC-5 and ARPE19 cells.*

- The reviewer is correct that it is difficult to distinguish the mode of viral spread in this type of infection assay. Thus rather than interpreting the experiment as evidence of blocking cell-to-cell dissemination, in the revised manuscript we present the experimental data as demonstration of viral inhibition throughout multiple cycles of infection.

- To further pursue this critical point, we have included Supplemental Figure 5 demonstrating that when the anti-gH mAbs 5C3 and 10C10 were added 24hpi for up to 6 days, we observed a reduced number of plaques and a decreased number of infected cells/plaque of AD169-IE2/YFP-infected fibroblasts. The data suggests that the mAbs block viral dissemination by limiting cell free infection and possibly cell-to-cell spread. However, without analyzing clinical CMV strains that exclusively spread by cell-to-cell transfer, we cannot exclude the possibility that the mAbs were blocking dissemination of AD169-IE2/YFP in a cell free manner. Thus, to not over interpret our data, we only discuss that the newly identified mAbs effectively limit virus dissemination.
15. *The text associated with D, E is very hard to follow (lines 193-202). It seems that reference is often made to the wrong panel.*
- We have reworded the text describing figures D and E to more clearly describe the data, and we have adjusted the data layout so that it is easier to understand.
16. *Fig 6 epitope mapping. The suggested epitope for 5C3 and 10C10 at 485-493 of gH exactly overlap with the epitope of 14-4b (and likely MSL-109) as suggested in Schultz 2016 JVI. This should be noted. And given that both 5C3 and 10C10 seem to have no new properties compared to those already described for 14-4b, how are these "novel"? The authors are encouraged to provide direct comparisons of their mAbs with 14-4b...esp in the competitive binding assays.*
- We thank the reviewer for bringing this important point to our attention. We have addressed the reviewers concerns with additional experiments presented as Supplemental Figure 8, which include several studies that compare the neutralizing efficacy, binding capacity gH mutants, and competition studies of 10C10, 5C3 and 14-4b. In general, the findings demonstrate that the three antibodies have different binding requirements to gH, even though they likely bind to a similar region within gH.
 - To address the epitope differences of MSL-109 with 5C3 and 10C10, data in Figure 8D and Table S4 demonstrate that MSL-109 binding is negatively impacted by the W168A gH mutant compared to 5C3 and 10C10. This mutant was found in a study (Fouts et al 2014 PNAS) generating MSL-109 escape viral mutants along with other mutations (P171 and D446). In addition, two studies using 2D-EM analysis of MSL-109 binding to the gH trimer identified residues 380-396 and 418-423 (Ciferri et al 2015 PNAS) and residues 403-419 and 442-446 (Ciferri et al 2015 PLoS Path). Thus, the exact epitope of MSL-109 has yet to be completely defined and quite distant from the predicted 5C3 and 10C10 binding site (aa 485-493). This point is addressed in the text.
 - Upon comparing the possible epitope of 14-4b to the newly identified gH mAbs 5C3 and 10C10 (Supplemental Figure 8), 14-4b has neutralizing capacity similar to 5C3, yet its binding characteristics are distinct from 5C3 and 10C10 (Supplemental Figure 8A &B). Schultz 2016 JVI demonstrated that either aa 485-493 and 671-675 of gH are important for binding to 14-4b suggesting that the epitope of 14-4b is not exactly defined in this study and is likely dependent on maintenance of the gH conformational. Using gH mutants, the 14-4b epitope was found to have a similar epitope profile to 10C10 (Supplemental Figure 8C), even though some 14-4b binding was observed for the 484-485 aa and none for 10C10. However, the competition studies show that 14-4b and 10C10 do not compete for the same binding site (Supplemental Figure 8D) suggesting that the two antibodies have distinct epitopes. On the other hand, the gH mutant binding studies demonstrate that 5C3 binding was quite different than 14-4b and 14-4b can only slightly limit 5C3 binding in the competition studies at only higher (>1.75X) antibody concentrations (Supplemental Figure 8C&D), possibly due to steric interference. While the 14-4b epitope does appear to be in close proximity to the epitopes of 5C3 and 10C10, the three antibodies have different binding properties to gH. Interestingly, the observation that 14-4b likely targets a region proximal to the epitope of 5C3 and 10C10 strongly supports the idea that there is an immunogenic

region of the gH protein that are susceptible to antibody-mediated neutralization. We believe that this discovery has far-reaching implications for vaccine design and antibody development. We have included this rationale in the revised discussion.

Reviewer 3 comments

In this interesting manuscript by Gardner and colleagues, there are significant and novel observations made about innovative methodologies to interrogate the humoral immune response to human cytomegalovirus infection. The over-arching theme is a focus on the so-called pentameric complex (PC) encoded by CMV. The key importance of the PC is that it mediates uptake of virus and subsequent infection in all cell types, including fibroblasts, epithelial cells, and myeloid cells. Hence, antibody to the PC is predicted to be broadly neutralizing to for all cell types, making it an ideal vaccine candidate for both active immunization and passive (antibody-mediated) immunization, such as might be administered to high-risk pregnant women with primary or recurrent CMV infection.

In this study, a novel robotics approach was taken, in combination with a high through-put neutralization assay, to screen for and subsequently identify a series of monoclonal antibodies target the CMV gH protein (an integral component of the pentameric complex that mediates cell entry). Detailed conformational analyses identified that the monoclonals identified bind to either the gH trimer formed with gH/gL/gO, or the gH pentamer formed by gH/gL/UL128/UL130/UL131. Key epitopes essential for viral entry are elucidated, and mechanisms of neutralization are described. One significant strength of the manuscript is that it goes beyond the descriptive nature of many similar studies to employ an informatics approach to rigorously define the relevant targets, including an alpha-helix rich domain that is targeted for neutralization.

Another strength of the manuscript is that the methodology described in the paper provides a framework for future discovery and design of vaccines and immune based therapies that will advance not only the CMV field, but antiviral immunologics for other pathogens. Hence, the manuscript will have broad interest to the readership, not only basic scientists and immunologists, but also vaccine designers, pediatricians and infectious diseases specialists, public health and policy experts, and anyone interested in childhood disabilities.

The manuscript in general is well-written and the experiments logically described. The conclusions are supported by the data and the controls are appropriate. A few areas for improvement are noted.

- 1. Lines 41-46. The authors should note that CMV immune globulin has not been definitively shown to protect the CMV-vulnerable fetus. Indeed, a controlled trial demonstrated no benefit (A randomized trial of hyperimmune globulin to prevent congenital cytomegalovirus. Revello MG, Lazzarotto T, Guerra B, Spinillo A, Ferrazzi E, Kustermann A, Guaschino S, Vergani P, Todros T, Frusca T, Arossa A, Furione M, Rognoni V, Rizzo N, Gabrielli L, Klersy C, Gerna G; CHIP Study Group. N Engl J Med. 2014;370(14):1316-26.). The authors should include this reference and may wish to add a comment on what the possible deficiencies were in the antibody preparation, and what improvements would be required (that the authors may be able to address) to improve the track record with immune globulin therapy in pregnancy.*
 - The reviewer brings up a very good point and we have included a discussion of the controlled trials of CMV immune globulin in the updated manuscript.*
- 2. In the screen, mice were immunized with AD169, precisely because it does not encode a functional pentameric complex (lines 68-70), and ~1500 candidate hybridomas were screened. A total of 6 highly neutralizing clones were subjected to further analysis. This demonstrated that mAbs 10C10 and 5C3 (group II) recovered proteins of ~100kDa, 37kDa, and 12kDa consistent with gH, gL and UL128 proteins. But it is unclear from the subsequent discussion (lines 90-105) whether the UL128 protein was recognized by this monoclonal. This point raised by the authors needs to be clarified. If this indeed was*

UL128, the authors should explain how this could be identified in AD169 virions, since UL128 is not supposed to be incorporated into virions in AD169 according to the work of other authors, since AD169 cannot generate a PC because of a mutation.

- The reviewer is correct in pointing out that AD169 virions do not incorporate an intact PC, however various reports have demonstrated that AD169 contains a truncated UL131, but UL128 and UL130 are still expressed in virus-infected cells in the ER. Thus we would expect to see these proteins in our biochemical analysis of infected cell lysates. We have included a brief discussion of this including references to pertinent literature in the revised manuscript.
3. *The gH monoclonals neutralized pentameric complex-containing clinical isolates more effectively than the gB monoclonals, compatible with a conformation-dependent targeting of gH in the pentamer. However, it is noted (line 129-129) that Cytogam was unable to block infectivity of these strains. This seems incompatible with the observations of Auerbach (reference 31) that the majority of the neutralizing activity in Cytogam targets the pentamer. The authors should reconcile this apparent discrepancy.*
- The 2012 paper by the Feirbach group referred to by the reviewer demonstrates that the major neutralizing antibody response by CMV hyperimmune globulin is directed toward the gH/gL/UL128/131 complex. However the authors demonstrate that while CMV IgG potently neutralizes CMV infection of endothelial and epithelial cells in vitro, it is not very effective at blocking infection of fibroblasts. The data presented in the paper, in fact, are directly in line with our own findings. In order to avoid confusion we have emphasized the parody between these studies in greater detail.
4. *A strength of the study is the use of the novel PEPperCHIP® assay, consisting of 217 overlapping cyclic peptides corresponding to gH. Interestingly, the 5C3 monoclonal bound reproducibly to a peptide, RREIFIVET corresponding to domain 2 of gH. A 3D model of gH based on the structure of HSV-2 gH predicted that aa481-493 fall in an alpha helix domain on the surface of gH. The authors should comment, based on review of the literature, if this epitope has been reported by other investigators mapping anti-gH response. Is this a novel, heretofore unrecognized epitope? If so, this would add to the novelty and value of the manuscript.*
- We thank the reviewer for the suggestion and have shown by additional studies that the epitope and binding of 10C10 and 5C3 are quite unique compared to other anti-gH antibodies MSL-109 and 14-4b. We believe that this study serves to emphasize the importance of this central alpha helix domain in CMV entry, and its potential as a target for potent immunotherapeutics.
5. *The fact that the humanized 5C3 monoclonal continued to show specificity in flow cytometry staining and AD169 neutralization (lines 247-255). However it is not clear if neutralization assays with the humanized monoclonal were performed with the humanized variant. This would be predicted, but the authors should comment on this point since it greatly enhances the therapeutic potential of the antibody. This information might be in the figure legend for figure S7 but it is unclear. This point needs to be elucidated and clarified.*
- We have included a brief discussion of the high level of conservation of the epitope region- including within clinically relevant strains. We agree that consideration of the ability of the humanized mAb to target these viruses greatly enhances the potential therapeutic value of these new tools.

REVIEWERS' COMMENTS:

Reviewer #1 (Remarks to the Author):

In this revised version, the authors have responded to review comments and made requested changes or justified why specific changes could not be made.

Reviewer #2 (Remarks to the Author):

The authors' have reasonably addressed this reviewer's concerns and questions.

Reviewer #3 (Remarks to the Author):

In this interesting revised manuscript by Gardner and colleagues, there are significant and novel observations made about innovative methodologies to interrogate the humoral immune response to human cytomegalovirus infection. The over-arching theme is a focus on the pentameric complex (PC) encoded by CMV. The key importance of the PC is that it mediates uptake of virus and subsequent infection in all cell types, including fibroblasts, epithelial cells, and myeloid cells. Hence, antibody to the PC is predicted to be broadly neutralizing to for all cell types, making it an ideal vaccine candidate for both active immunization and passive (antibody-mediated) immunization, such as might be administered to high-risk pregnant women with primary or recurrent CMV infection.

In this study, a novel robotics approach was taken, in combination with a high through-put neutralization assay, to screen for and subsequently identify a series of monoclonal antibodies target the CMV gH protein (an integral component of the pentameric complex that mediates cell entry). Detailed conformational analyses identified that the monoclonals identified bind to either the gH trimer formed with gH/gL/gO, or the gH pentamer formed by gH/gL/UL128/UL130/UL131. Key epitopes essential for viral entry are elucidated, and mechanisms of neutralization are described. One significant strength of the manuscript is that it goes beyond the descriptive nature of many similar studies to employ an informatics approach to rigorously define the relevant targets, including an alpha-helix rich domain that is targeted for neutralization.

Another strength of the manuscript is that the methodology described in the paper provides a framework for future discovery and design of vaccines and immune based therapies that will advance not only the CMV field, but antiviral immunologics for other pathogens. Hence, the manuscript will have broad interest to the readership, not only basic scientists and immunologists, but also vaccine designers, pediatricians and infectious diseases specialists, public health and policy experts, and anyone interested in childhood disabilities.

The revised manuscript is well-written and the experiments logically described. The conclusions are supported by the data and the controls are appropriate. The authors have been highly responsive to previous reviews. Specific improvements in response to previous

reviews include:

1. The authors have included an updated discussion of the controlled trials of CMV immune globulin in the updated manuscript.
2. The authors have improved the discussion of the truncation of UL131, and the rationale for finding UL128 and UL130 which are still expressed in virus-infected cells in the ER in the AD169 virus.
3. The authors have improved the clarity of the discussion of their observations which are clearly discussed in the context of work of other investigators.
4. There is also an improved discussion of neutralization of clinical isolates.

Overall, the application is highly responsive to previous reviews and the manuscript will represent a valuable contribution to the existing literature on this topic.